# BRAIN DECODING: TOWARD REAL-TIME RECONSTRUCTION OF VISUAL PERCEPTION

**Yohann Benchetrit**[1,*]**, Hubert Banville**[1,*]**, Jean-Rémi King**[1,2]
[1]FAIR, Meta, [2] Laboratoire des Systèmes Perceptifs, École Normale Supérieure, PSL University
{ybenchetrit,hubertjb,jeanremi}@meta.com

## ABSTRACT

In the past five years, the use of generative and foundational AI systems has greatly improved the decoding of brain activity. Visual perception, in particular, can now be decoded from functional Magnetic Resonance Imaging (fMRI) with remarkable fidelity. This neuroimaging technique, however, suffers from a limited temporal resolution ($\approx$0.5 Hz) and thus fundamentally constrains its real-time usage. Here, we propose an alternative approach based on magnetoencephalography (MEG), a neuroimaging device capable of measuring brain activity with high temporal resolution ($\approx$5,000 Hz). For this, we develop an MEG decoding model trained with both contrastive and regression objectives and consisting of three modules: i) pretrained embeddings obtained from the image, ii) an MEG module trained end-to-end and iii) a pretrained image generator. Our results are threefold: Firstly, our MEG decoder shows a 7X improvement of image-retrieval over classic linear decoders. Second, late brain responses to images are best decoded with DINOv2, a recent foundational image model. Third, image retrievals and generations both suggest that high-level visual features can be decoded from MEG signals, although the same approach applied to 7T fMRI also recovers better low-level features. Overall, these results, while preliminary, provide an important step towards the decoding – in real-time – of the visual processes continuously unfolding within the human brain.

## 1 INTRODUCTION

**Automating the discovery of brain representations.** Understanding how the human brain represents the world is arguably one of the most profound scientific challenges. This quest, which originally consisted of searching, one by one, for the specific features that trigger each neuron, (*e.g.* Hubel & Wiesel (1962); O'Keefe & Nadel (1979); Kanwisher et al. (1997)), is now being automated by Machine Learning (ML) in two main ways. First, as a signal processing *tool*, ML algorithms are trained to extract informative patterns of brain activity in a data-driven manner. For example, Kamitani & Tong (2005) trained a support vector machine to classify the orientations of visual gratings from functional Magnetic Resonance Imaging (fMRI). Since then, deep learning has been increasingly used to discover such brain activity patterns (Roy et al., 2019; Thomas et al., 2022; Jayaram & Barachant, 2018; Défossez et al., 2022; Scotti et al., 2023). Second, ML algorithms are used as functional *models* of the brain. For example, Yamins et al. (2014) have shown that the embedding of natural images in pretrained deep nets linearly account for the neuronal responses to these images in the cortex. Since, pretrained deep learning models have been shown to account for a wide variety of stimuli including text, speech, navigation, and motor movement (Banino et al., 2018; Schrimpf et al., 2020; Hausmann et al., 2021; Mehrer et al., 2021; Caucheteux et al., 2023).

**Generating images from brain activity.** This observed representational alignment between brain activity and deep learning models creates a new opportunity: decoding of visual stimuli need not be restricted to a limited set of classes, but can now leverage pretrained representations to condition subsequent generative AI models. While the resulting image may be partly "hallucinated", interpreting images can be much simpler than interpreting latent features. Following a long series

---

*Equal contribution.

of generative approaches (Nishimoto et al., 2011; Kamitani & Tong, 2005; VanRullen & Reddy, 2019; Seeliger et al., 2018), diffusion techniques have, in this regard, significantly improved the generation of images from functional Magnetic Resonance Imaging (fMRI). The resulting pipeline typically consists of three main modules: (1) a set of pretrained embeddings obtained from the image onto which (2) fMRI activity can be linearly mapped and (3) ultimately used to condition a pretrained image-generation model (Ozcelik & VanRullen, 2023; Mai & Zhang, 2023; Zeng et al., 2023; Ferrante et al., 2022). These recent fMRI studies primarily differ in the type of pretrained image-generation model that they use.

**The challenge of real-time decoding.** This generative decoding approach has been mainly applied to fMRI. However, the temporal resolution of fMRI is limited by the time scale of blood flow and typically leads to one snapshot of brain activity every two seconds – a time scale that challenges its clinical usage, *e.g.* for patients who require a brain-computer-interface (Willett et al., 2023; Moses et al., 2021; Metzger et al., 2023; Défossez et al., 2022). On the contrary, magnetoencephalography (MEG) can measure brain activity at a much higher temporal resolution ($\approx$5,000 Hz) by recording the fluctuation of magnetic fields elicited by the post-synaptic potentials of pyramidal neurons. This higher temporal resolution comes at a cost, however: the spatial resolution of MEG is limited to $\approx$300 sensors, whereas fMRI measures $\approx$100,000 voxels. In sum, fMRI intrinsically limits our ability to (1) track the dynamics of neuronal activity, (2) decode dynamic stimuli (speech, videos, etc.) and (3) apply these tools to real-time use cases. Conversely, it is unknown whether temporally-resolved neuroimaging systems like MEG are sufficiently precise to generate natural images in real-time.

**Our approach.** Combining previous work on speech retrieval from MEG (Défossez et al., 2022) and on image generation from fMRI (Takagi & Nishimoto, 2023; Ozcelik & VanRullen, 2023), we here develop a three-module pipeline trained to align MEG activity onto pretrained visual embeddings and generate images from a stream of MEG signals (Fig. 1).

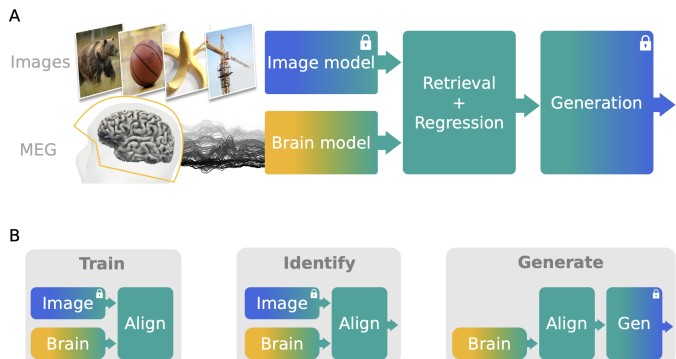

Figure 1: (**A**) Approach. Locks indicate pretrained models. (**B**) Processing schemes. Unlike image generation, retrieval happens in latent space, but requires the true image in the retrieval set.

Our approach provides three main contributions: our MEG decoder (1) yields a 7X increase in performance as compared to linear baselines (Fig. 2), (2) helps reveal when high-level semantic features are processed in the brain (Fig. 3) and (3) allows the continuous generation of images from temporally-resolved brain signals (Fig. 4). Overall, this approach thus paves the way to better understand the unfolding of the brain responses to visual inputs.

## 2 METHODS

### 2.1 PROBLEM STATEMENT

We aim to decode images from multivariate time series of brain activity recorded with MEG as healthy participants watched a sequence of natural images. Let $\boldsymbol{X}_i \in \mathbb{R}^{C \times T}$ be the MEG time window collected as an image $I_i$ was presented to the participant, where $C$ is the number of MEG

channels, $T$ is the number of time points in the MEG window and $i \in [\![1, N]\!]$, with $N$ the total number of images. Let $\boldsymbol{z}_i \in \mathbb{R}^F$ be the latent representation of $I_i$, with $F$ the number of features, obtained by embedding the image using a pretrained image model (Section 2.4). As described in more detail below, our decoding approach relies on training a *brain module* $\mathbf{f}_\theta : \mathbb{R}^{C \times T} \to \mathbb{R}^F$ to maximally retrieve or predict $I_i$ through $\boldsymbol{z}_i$, given $\boldsymbol{X}_i$.

## 2.2 TRAINING OBJECTIVES

We use different training objectives for the different parts of our proposed pipeline. First, in the case of retrieval, we aim to pick the right image $I_i$ (*i.e.*, the one corresponding to $\boldsymbol{X}_i$) out of a bank of candidate images. To do so, we train $\mathbf{f}_\theta$ using the CLIP loss (Radford et al., 2021) (*i.e.*, the InfoNCE loss (Oord et al., 2018) applied in both brain-to-image and image-to-brain directions) on batches of size B with exactly one positive example,

$$\mathcal{L}_{CLIP}(\theta) = -\frac{1}{B} \sum_{i=1}^{B} \left( \log \frac{\exp(s(\hat{\boldsymbol{z}}_i, \boldsymbol{z}_i)/\tau)}{\sum_{j=1}^{B} \exp(s(\hat{\boldsymbol{z}}_i, \boldsymbol{z}_j)/\tau)} + \log \frac{\exp(s(\hat{\boldsymbol{z}}_i, \boldsymbol{z}_i)/\tau)}{\sum_{k=1}^{B} \exp(s(\hat{\boldsymbol{z}}_k, \boldsymbol{z}_i)/\tau)} \right) \quad (1)$$

where $s$ is the cosine similarity, $\boldsymbol{z}_i$ and $\hat{\boldsymbol{z}}_i = \mathbf{f}_\theta(\boldsymbol{X}_i)$ are the latent representation and the corresponding MEG-based prediction, respectively, and $\tau$ is a learned temperature parameter.

Next, to go beyond retrieval and instead generate images, we train $\mathbf{f}_\theta$ to directly predict the latent representations $\boldsymbol{z}$ such that we can use them to condition generative image models. This is done using a standard mean squared error (MSE) loss over the (unnormalized) $\boldsymbol{z}_i$ and $\hat{\boldsymbol{z}}_i$:

$$\mathcal{L}_{MSE}(\theta) = \frac{1}{NF} \sum_{i=1}^{N} \|\boldsymbol{z}_i - \hat{\boldsymbol{z}}_i\|_2^2 \quad (2)$$

Finally, we combine the CLIP and MSE losses using a convex combination with tuned weight to train models that benefit from both training objectives:

$$\mathcal{L}_{Combined} = \lambda \mathcal{L}_{CLIP} + (1 - \lambda)\mathcal{L}_{MSE} \quad (3)$$

## 2.3 BRAIN MODULE

We adapt the dilated residual ConvNet architecture of Défossez et al. (2022), denoted as $\mathbf{f}_\theta$, to learn the projection from an MEG window $\boldsymbol{X}_i \in \mathbb{R}^{C \times T}$ to a latent image representation $\boldsymbol{z}_i \in \mathbb{R}^F$. The original model's output $\hat{\boldsymbol{Y}}_{backbone} \in \mathbb{R}^{F' \times T}$ maintains the temporal dimension of the network through its residual blocks. However, here we regress a single latent per input instead of a sequence of $T$ latents like in Défossez et al. (2022). Consequently, we add a temporal aggregation layer to reduce the temporal dimension of $\hat{\boldsymbol{Y}}_{backbone}$ to obtain $\hat{\boldsymbol{y}}_{agg} \in \mathbb{R}^{F'}$. We experiment with three types of aggregations: global average pooling, a learned affine projection, and an attention layer. Finally, we add two MLP heads, *i.e.*, one for each term in $\mathcal{L}_{Combined}$, to project from $F'$ to the $F$ dimensions of the target latent. Additional details on the architecture can be found in Appendix A.1.

We run a hyperparameter search to identify an appropriate configuration of preprocessing, brain module architecture, optimizer and CLIP loss hyperparameters for the retrieval task (Appendix A.2). The final architecture configuration for retrieval is described in Table S1 and contains *e.g.* 6.4M trainable parameters for $F = 768$. The final architecture uses two convolutional blocks and an affine projection to perform temporal aggregation (further examined in Appendix A.11).

For image generation experiments, the output of the MSE head is further postprocessed as in Ozcelik & VanRullen (2023), *i.e.*, we z-score normalize each feature across predictions, and then apply the inverse z-score transform fitted on the training set (defined by the mean and standard deviation of each feature dimension on the target embeddings). We select $\lambda$ in $\mathcal{L}_{Combined}$ by sweeping over $\{0.0, 0.25, 0.5, 0.75\}$ and pick the model whose top-5 accuracy is the highest on the "large test set" (which is disjoint from the "small test set" used for generation experiments; see Section 2.8). When training models to generate CLIP and AutoKL latents, we simplify the task of the CLIP head

by reducing the dimensionality of its target: we use the CLS token for CLIP-Vision ($F_{MSE} = 768$), the "mean" token for CLIP-Text ($F_{MSE} = 768$), and the channel-average for AutoKL latents ($F_{MSE} = 4096$), respectively.

Of note, when comparing performance on different window configurations *e.g.* to study the dynamics of visual processing in the brain, we train a different model per window configuration. Despite receiving a different window of MEG as input, these models use the same latent representations of the corresponding images.

## 2.4 IMAGE MODULES

We study the functional alignment between brain activity and a variety of (output) embeddings obtained from deep neural networks trained in three different representation learning paradigms, spanning a wide range of dimensionalities: supervised learning (VGG-19), image-text alignment (CLIP), and variational autoencoders. When using vision transformers, we further include two additional embeddings of smaller dimensionality: the average of all output embeddings across tokens (mean), and the output embedding of the class-token (CLS). For comparison, we also evaluate our approach on human-engineered features obtained without deep learning. The list of embeddings is provided in Appendix A.3. For clarity, we focus our experiments on a representative subset.

## 2.5 GENERATION MODULE

To fairly compare our work to the results obtained with fMRI results, we follow the approach of Ozcelik & VanRullen (2023) and use a model trained to generate images from pretrained embeddings. Specifically, we use a latent diffusion model conditioned on three embeddings: CLIP-Vision (257 tokens $\times$ 768), CLIP-Text (77 tokens $\times$ 768), and a variational autoencoder latent (AutoKL; ($4 \times 64 \times 64$). In particular, we use the CLIP-Text embeddings obtained from the THINGS object-category of a stimulus image. Following Ozcelik & VanRullen (2023), we apply diffusion with 50 DDIM steps, a guidance of 7.5, a strength of 0.75 with respect to the image-to-image pipeline, and a mixing of 0.4.

## 2.6 TRAINING AND COMPUTATIONAL CONSIDERATIONS

Cross-participant models are trained on a set of $\approx$63,000 examples using the Adam optimizer (Kingma & Ba, 2014) with default parameters ($\beta_1$=0.9, $\beta_2$=0.999), a learning rate of $3 \times 10^{-4}$ and a batch size of 128. We use early stopping on a validation set of $\approx$15,800 examples randomly sampled from the original training set, with a patience of 10, and evaluate the performance of the model on a held-out test set (see below). Models are trained on a single Volta GPU with 32 GB of memory. We train each model three times using three different random seeds for the weight initialization of the brain module.

## 2.7 EVALUATION

**Retrieval metrics.** We first evaluate decoding performance using retrieval metrics. For a known test set, we are interested in the probability of identifying the correct image given the model predictions. Retrieval metrics have the advantage of sharing the same scale regardless of the dimensionality of the MEG (like encoding metrics) or the dimensionality of the image embedding (like regression metrics). We evaluate retrieval using either the *relative median rank* (which does not depend on the size of the retrieval set), defined as the rank of a prediction divided by the size of the retrieval set, or the *top-5 accuracy* (which is more common in the literature). In both cases, we use cosine similarity to evaluate the strength of similarity between feature representations (Radford et al., 2021).

**Generation metrics.** Decoding performance is often measured qualitatively as well as quantitatively using a variety of metrics reflecting the reconstruction fidelity both in terms of perception and semantics. For fair comparison with fMRI generations, we provide the same metrics as Ozcelik & VanRullen (2023), computed between seen and generated images: PixCorr (the pixel-wise correlation between the true and generated images), SSIM (Structural Similarity Index Metric), and SwAV (the correlation with respect to SwAV-ResNet50 output). On the other hand, AlexNet(2/5), Inception, and CLIP are the respective 2-way comparison scores of layers 2/5 of AlexNet, the pooled last

layer of Inception and the output layer of CLIP. For the NSD dataset, these metrics are reported for participant 1 only (see Appendix A.4).

To avoid non-representative cherry-picking, we sort all generations on the test set according to the sum of (minus) SwAV and SSIM. We then split the data into 15 blocks and pick 4 images from the best, middle and worst blocks with respect to the summed metric (Figures S2 and S5).

**Real-time and average metrics.**   It is common in fMRI to decode brain activity from preprocessed values estimated with a General Linear Model. These "beta values" are estimates of brain responses to individual images, computed across multiple repetitions of such images. To provide a fair assessment of possible MEG decoding performance, we thus leverage repeated image presentations available in the datasets (see below) by averaging predictions before evaluating metrics and generating images.

## 2.8   DATASET

We test our approach on the THINGS-MEG dataset (Hebart et al., 2023). Four participants (2 female, 2 male; mean age of 23.25 years), underwent 12 MEG sessions during which they were presented with a set of 22,448 unique images selected from the THINGS database (Hebart et al., 2019), covering 1,854 categories. Of those, only a subset of 200 images (each one of a different category) was shown multiple times to the participants. The images were displayed for 500 ms each, with a variable fixation period of $1000\pm200$ ms between presentations. The THINGS dataset additionally contains 3,659 images that were not shown to the participants and that we use to augment the size of our retrieval set and emphasize the robustness of our method.

**MEG preprocessing.**   We use a minimal MEG data-preprocessing pipeline as in Défossez et al. (2022). Raw data from the 272 MEG radial gradiometer channels is downsampled from 1,200 Hz to 120 Hz. The continuous MEG data is then epoched from -500 ms to 1,000 ms relative to stimulus onset and baseline-corrected by subtracting the mean signal value observed between the start of an epoch and the stimulus onset for each channel. Finally, we apply a channel-wise robust scaler (Pedregosa et al., 2011) and clip values outside of $[-20, 20]$ to minimize the impact of large outliers.

**Splits.**   The original split of Hebart et al. (2023) consists of 22,248 uniquely presented images, and 200 test images repeated 12 times each for each participant (*i.e.*, 2,400 trials per participant). The use of this data split presents a challenge, however, as the test set contains only one image per category, and these categories are also seen in the training set. This means evaluating retrieval performance on this test set does not measure the capacity of the model to (1) extrapolate to new unseen categories of images and (2) recover a particular image within a set of multiple images of the same category, but rather only to "categorize" it. Consequently, we propose two modifications of the original split. First, we remove from the training set any image whose category appears in the original test set. This "adapted training set" removes any categorical leakage across the train/test split and makes it possible to assess the capacity of the model to decode images of unseen image categories (*i.e.*, a "zero-shot" setting). Second, we propose a new "large test set" that is built using the images removed from the training set. This new test set effectively allows evaluating retrieval performance of images within images of the same category[1]. We report results on both the original ("small") and the "large" test sets to enable comparisons with the original settings of Hebart et al. (2023). Finally, we also compare our results to the performance obtained by a similar pipeline but trained on fMRI data using the NSD dataset (Allen et al., 2022) (see Appendix A.4).

## 3   RESULTS

**ML as an effective *model* of the brain.**   Which representations of natural images are likely to maximize decoding performance? To answer this question, we compare the retrieval performance obtained by linear Ridge regression models trained to predict one of 16 different latent visual representations given the flattened MEG response $X_i$ to each image $I_i$ (see Appendix A.5 and black

---

[1]We leave out images of the original test set from this new large test set, as keeping them would create a discrepancy between the number of MEG repetitions for training images and test images.

transparent bars in Fig. 2). While all image embeddings lead to above-chance retrieval, supervised and text/image alignment models (*e.g.* VGG, CLIP) yield the highest retrieval scores.

**ML as an effective *tool* to learn brain responses.** We then compare these linear baselines to a deep ConvNet architecture (Défossez et al., 2022) trained on the same dataset to retrieve the matching image given an MEG window[2]. Using a deep model leads to a 7X improvement over the linear baselines (Fig. 2). Multiple types of image embeddings lead to good retrieval performance, with VGG-19 (supervised learning), CLIP-Vision (text/image alignment) and DINOv2 (self-supervised learning) yielding top-5 accuracies of 70.33±2.80%, 68.66±2.84%, 68.00±2.86%, respectively (where the standard error of the mean is computed across the averaged image-wise metrics). Similar conclusions, although with lower performance, can be drawn from our "large" test set setting, where decoding cannot rely solely on the image category but also requires discriminating between multiple images of the same category. Representative retrieval examples are shown in Appendix A.7.

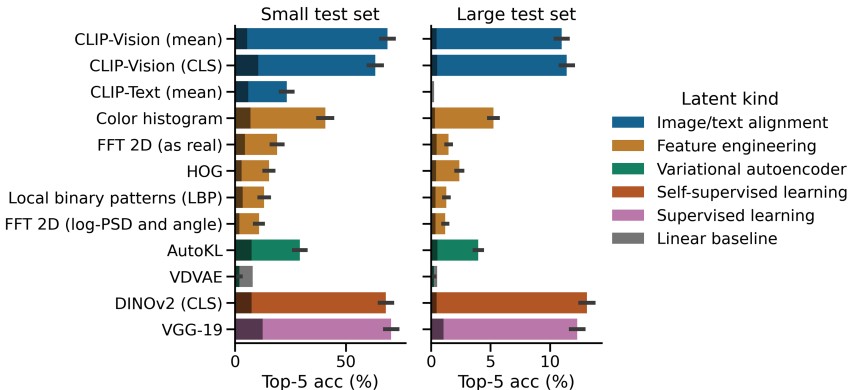

Figure 2: Image retrieval performance obtained from a trained deep ConvNet. Linear decoder baseline performance (see Table S2) is shown with a black transparent bar for each latent. The original "small" test set (Hebart et al., 2023) comprises 200 distinct images, each belonging to a different category. In contrast, our proposed "large" test set comprises 12 images from each of those 200 categories, yielding a total of 2,400 images. Chance-level is 2.5% top-5 accuracy for the small test set and 0.21% for the large test set. The best latent representations yield accuracies around 70% and 13% for the small and large test sets, respectively.

**Temporally-resolved image retrieval.** The above results are obtained from the full time window (-500 to 1,000 ms relative to stimulus onset). To further investigate the feasibility of decoding visual representations as they unfold in the brain, we repeat this analysis on 100-ms sliding windows with a stride of 25 ms (Fig. 3). For clarity, we focus on a subset of representative image embeddings. As expected, all models yield chance-level performance before image presentation. For all embeddings, a first clear peak can be observed for windows ending around 200-275 ms after image onset. A second peak follows for windows ending around 150-200 ms after image offset. Supplementary analysis (Fig. S7) further suggests these two peak intervals contain complementary information for the retrieval task. Finally, performance quickly goes back to chance-level. Interestingly, the recent self-supervised model DINOv2 yields particularly high retrieval performance after image offset.

Representative time-resolved retrieval examples are shown in Appendix A.7. Overall, the retrieved images tend to come from the correct category, such as "speaker" or "brocoli", mostly during the first few sub-windows ($t \leq 1$ s). However, these retrieved images do not appear to share obvious low-level features to the images seen by the participants.

While further analyses of these results remain necessary, it seems that (1) our decoding leverages the brain responses related to both the onset and the offset of the image and (2) category-level information dominates these visual representations as early as 250 ms.

---

[2]We use $\lambda = 1$ in $\mathcal{L}_{Combined}$ as we are solely concerned with the retrieval part of the pipeline here.

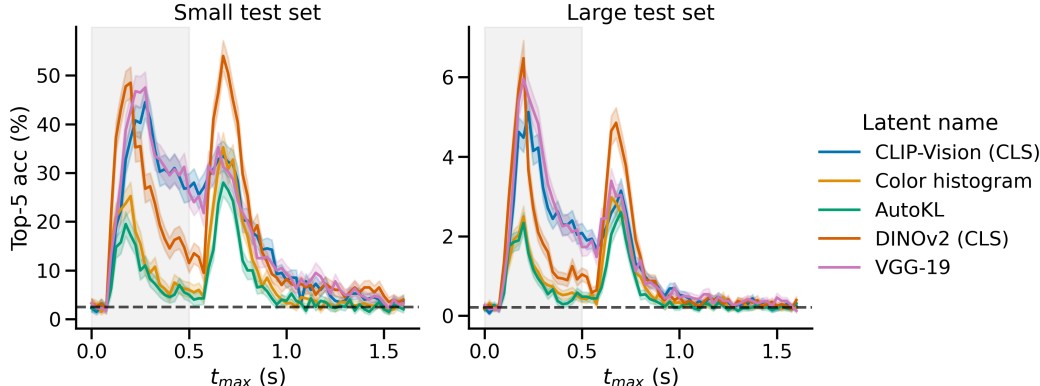

Figure 3: Retrieval performance of models trained on 100-ms sliding windows with a stride of 25 ms for different image representations. The shaded gray area indicates the 500-ms interval during which images were presented to the participants and the horizontal dashed line indicates chance-level performance. Accuracy peaks a few hundreds of milliseconds after both the image onset and offset for all embeddings.

Table 1: Quantitative evaluation of reconstruction quality from MEG data on THINGS-MEG (compared to fMRI data on NSD (Allen et al., 2022) using a cross-validated Ridge regression). We report PixCorr, SSIM, AlexNet(2), AlexNet(5), Inception, SwAV and CLIP and their SEM when meaningful. In particular, this shows that fMRI betas as provided in NSD are significantly easier to decode than MEG signals from THINGS-MEG.

| Dataset | Low-level | | | High-level | | | |
|---|---|---|---|---|---|---|---|
| | PixCorr ↑ | SSIM ↑ | AlexNet(2) ↑ | AlexNet(5) ↑ | Inception ↑ | CLIP ↑ | SwAV ↓ |
| NSD (fMRI) | $0.305 \pm 0.007$ | $0.366 \pm 0.005$ | 0.962 | 0.977 | 0.910 | 0.917 | $0.410 \pm 0.004$ |
| THINGS-MEG (averaged across all trials within subject) | $0.076 \pm 0.005$ | $0.336 \pm 0.007$ | 0.736 | 0.826 | 0.671 | 0.767 | $0.584 \pm 0.004$ |
| THINGS-MEG (averaged across all trials and subjects) | $0.090 \pm 0.009$ | $0.341 \pm 0.015$ | 0.774 | 0.876 | 0.703 | 0.811 | $0.567 \pm 0.008$ |
| THINGS-MEG (no average) | $0.058 \pm 0.011$ | $0.327 \pm 0.014$ | 0.695 | 0.753 | 0.593 | 0.700 | $0.630 \pm 0.007$ |

**Generating images from MEG.** While framing decoding as a retrieval task yields promising results, it requires the true image to be in the retrieval set – a well-posed problem which presents limited use-cases in practice. To address this issue, we trained three distinct brain modules to predict the three embeddings that we use (see Section 2.5) to generate images. Fig. 4 shows example generations from (A) "growing" windows, *i.e.*, where increasingly larger MEG windows (from [0, 100] to [0, 1,500] ms after onset with 50 ms increments) are used to condition image generation and (B) full-length windows (*i.e.*, -500 to 1,000 ms). Additional full-window representative generation examples are shown in Appendix A.8. As confirmed by the evaluation metrics of Table 1 (see Table S4 for participant-wise metrics), many generated images preserve the high-level category of the true image. However, most generations appear to preserve a relatively small amount of low-level features, such as the position and color of each object. Lastly, we provide a sliding window analysis of these metrics in Appendix A.12. These results suggest that early responses to both image onset and offset are primarily associated with low-level metrics, while high-level features appear more related to brain activity in the 200-500 ms interval.

The application of a very similar pipeline on an analogous fMRI dataset (Allen et al., 2022; Ozcelik & VanRullen, 2023) – using a simple Ridge regression – shows image reconstructions that share both high-level and low-level features with the true image (Fig. S2). Together, these results suggest that it is not the reconstruction pipeline which fails to reconstruct low-level features, but rather the MEG signals which are comparatively harder to decode.

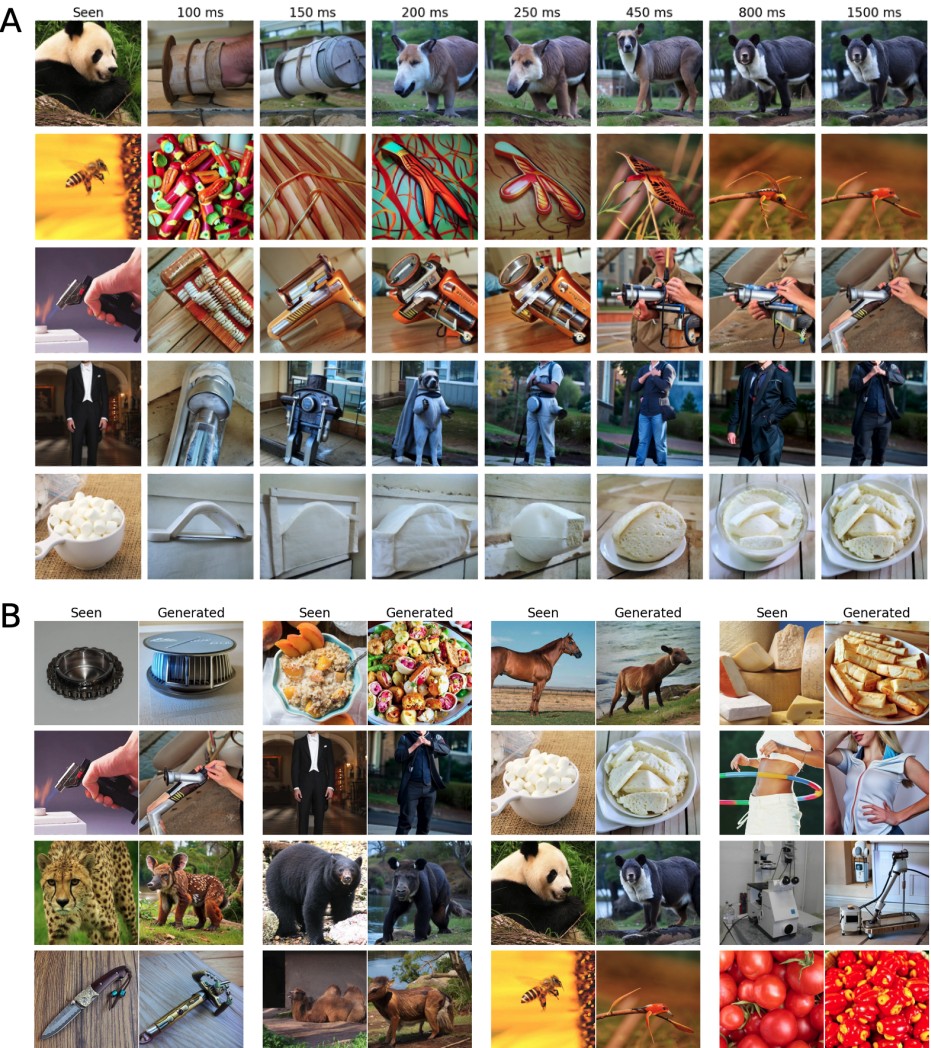

Figure 4: Handpicked examples of successful generations. (**A**) Generations obtained on growing windows starting at image onset (0 ms) and ending at the specified time. (**B**) Full-window generations (-500 to 1,000 ms).

## 4  DISCUSSION

**Related work.** The present study shares several elements with previous MEG and electroencephalography (EEG) studies designed not to maximize decoding performance but to understand the cascade of visual processes in the brain. In particular, previous studies have trained linear models to either (1) classify a small set of images from brain activity (Grootswagers et al., 2019; King & Wyart, 2021), (2) predict brain activity from the latent representations of the images (Cichy et al., 2017) or (3) quantify the similarity between these two modalities with representational similarity analysis (RSA) (Cichy et al., 2017; Bankson et al., 2018; Grootswagers et al., 2019; Gifford et al., 2022). While these studies also make use of image embeddings, their linear decoders are limited to classifying a small set of object classes, or to distinguishing pairs of images.

In addition, several deep neural networks have been introduced to maximize the classification of speech (Défossez et al., 2022), mental load (Jiao et al., 2018) and images (Palazzo et al., 2020; McCartney et al., 2022; Bagchi & Bathula, 2022) from EEG recordings. In particular, Palazzo et al. (2020) introduced a deep convolutional neural network to classify natural images from EEG signals. However, the experimental protocol consisted of presenting all of the images of the same class within

a single continuous block, which risks allowing the decoder to rely on autocorrelated noise, rather than informative brain activity patterns (Li et al., 2020). In any case, these EEG studies focus on the categorization of a relatively small number of images classes.

In sum, there is, to our knowledge, no MEG decoding study that learns end-to-end to reliably generate an open set of images.

**Impact.** Our methodological contribution has both fundamental and practical impacts. First, the decoding of perceptual representations could clarify the unfolding of visual processing in the brain. While there is considerable work on this issue, neural representations are challenging to interpret because they represent latent, abstract, feature spaces. Generative decoding, on the contrary, can provide concrete and, thus, interpretable predictions. Put simply, generating images at each time step could help neuroscientists understand whether specific – potentially unanticipated – textures or object parts are represented. For example, Cheng et al. (2023) showed that generative decoding applied to fMRI can be used to decode the subjective perception of visual illusions. Such techniques can thus help to clarify the neural bases of subjective perception and to dissociate them from those responsible for "copying" sensory inputs. Our work shows that this endeavor could now be applied to clarify *when* these subjective representations arise. Second, generative brain decoding has concrete applications. For example, it has been used in conjunction with encoding, to identify stimuli that maximize brain activity (Bashivan et al., 2019). Furthermore, non-invasive brain-computer interfaces (BCI) have been long-awaited by patients with communication challenges related to brain lesions. BCI, however, requires real-time decoding, and thus limits the use of neuroimaging modalities with low temporal resolution such as fMRI. This application direction, however, will likely require extending our work to EEG, which provides similar temporal resolution to MEG, but is typically much more common in clinical settings.

**Limitations.** Our analyses highlight three main limitations to the decoding of images from MEG signals. First, generating images from MEG appears worse at preserving low-level features than a similar pipeline on 7T fMRI (Fig. S2). This result resonates with the fact that the spatial resolution of MEG ($\approx$ cm) is much lower than 7T fMRI's ($\approx$ mm). Moreover, and consistent with previous findings (Cichy et al., 2014; Hebart et al., 2023), the low-level features can be predominantly extracted from the brief time windows immediately surrounding the onset and offset of brain responses. As a result, these transient low-level features might have a lesser impact on image generation compared to the more persistent high-level features. Second, the present approach directly depends on the pretraining of several models, and only learns end-to-end to align the MEG signals to these pretrained embeddings. Our results show that this approach leads to better performance than classical computer vision features such as color histograms, Fast Fourier transform and histogram of oriented gradients (HOG). This is consistent with a recent MEG study by Défossez et al. (2022) which showed, in the context of speech decoding, that pretrained embeddings outperformed a fully end-to-end approach. Nevertheless, it remains to be tested whether (1) fine-tuning the image and generation modules and (2) combining the different types of visual features could improve decoding performance.

**Ethical implications.** While the decoding of brain activity promises to help a variety of brain-lesioned patients (Metzger et al., 2023; Moses et al., 2021; Défossez et al., 2022; Liu et al., 2023; Willett et al., 2023), the rapid advances of this technology raise several ethical considerations, and most notably, the necessity to preserve mental privacy. Several empirical findings are relevant to this issue. Firstly, the decoding performance obtained with non-invasive recordings is only high for *perceptual* tasks. By contrast, decoding accuracy considerably diminishes when individuals are tasked to imagine representations (Horikawa & Kamitani, 2017; Tang et al., 2023). Second, decoding performance seems to be severely compromised when participants are engaged in disruptive tasks, such as counting backward (Tang et al., 2023). In other words, the subjects' consent is not only a legal but also and primarily a technical requirement for brain decoding. To delve into these issues effectively, we endorse the open and peer-reviewed research standards.

**Conclusion.** Overall, these results provide an important step towards the decoding of the visual processes continuously unfolding in the human brain.

ACKNOWLEDGMENTS

This work was funded in part by FrontCog grant ANR-17-EURE-0017 to JRK for his work at PSL.

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
