# A APPENDIX

## A.1 ADDITIONAL DETAILS ON THE BRAIN MODULE ARCHITECTURE

We provide additional details on the brain module $\mathbf{f}_\theta$ described in Section 2.3.

The brain module first applies two successive linear transformations in the spatial dimension to an input MEG window. The first linear transformation is the output of an attention layer conditioned on the MEG sensor positions. The second linear transformation is learned subject-wise, such that each subject ends up with their own linear projection matrix $\boldsymbol{W}_s^{subj} \in \mathbb{R}^{C \times C}$, with $C$ the number of input MEG channels and $s \in [\![1, S]\!]$ where $S$ is the number of subjects. The module then applies a succession of 1D convolutional blocks that operate in the temporal dimension and treat the spatial dimension as features. These blocks each contain three convolutional layers (dilated kernel size of 3, stride of 1) with residual skip connections. The first two layers of each block use GELU activations while the last one use a GLU activation. The output of the last convolutional block is passed through a learned linear projection to yield a different number of features $F'$ (fixed to 2048 in our experiments).

The resulting features are then fed to a temporal aggregation layer which reduces the remaining temporal dimension. Given the output of the brain module backbone $\hat{Y}_{backbone} \in \mathbb{R}^{F' \times T}$, we compare three approaches to reduce the temporal dimension of size $T$: (1) Global average pooling, *i.e.*, the features are averaged across time steps; (2) Learned affine projection in which the temporal dimension is projected from $\mathbb{R}^T$ to $\mathbb{R}$ using a learned weight vector $\boldsymbol{w}^{agg} \in \mathbb{R}^T$ and bias $b^{agg} \in \mathbb{R}$; (3) Bahdanau attention layer (Bahdanau et al., 2014) which predicts an affine projection from $\mathbb{R}^T$ to $\mathbb{R}$ conditioned on the input $\hat{Y}_{backbone}$ itself. Following the hyperparameter search of Appendix A.2, we selected the learned affine projection approach for our experiments. Finally, the resulting output is fed to CLIP and MSE head-specific MLP projection heads where a head consists of repeated LayerNorm-GELU-Linear blocks, to project from $F'$ to the $F$ dimensions of the target latent.

We refer the interested reader to Défossez et al. (2022) for a description of the original architecture, and to the code available at https://github.com/facebookresearch/brainmagick.

## A.2 HYPERPARAMETER SEARCH

We run a hyperparameter grid search to find an appropriate configuration (MEG preprocessing, optimizer, brain module architecture and CLIP loss) for the MEG-to-image retrieval task. We randomly split the 79,392 (MEG, image) pairs of the adapted training set (Section 2.8) into 60%-20%-20% train, valid and test splits such that all presentations of a given image are contained in the same split. We use the validation split to perform early stopping and the test split to evaluate the performance of a configuration.

For the purpose of this search we pick CLIP-Vision (CLS) latent as a representative latent, since it achieved good retrieval performance in preliminary experiments. We focus the search on the retrieval task, *i.e.*, by setting $\lambda = 1$ in Eq. 3, and leave the selection of an optimal $\lambda$ to a model-specific sweep using a held-out set (see Section 2.3). We run the search six times using two different random seed initializations for the brain module and three different random train/valid/test splits. Fig. S1 summarizes the results of this hyperparameter search.

Based on this search, we use the following configuration: MEG window $(t_{min}, t_{max})$ of $[-0.5, 1.0]$ s, learning rate of $3 \times 10^{-4}$, batch size of 128, brain module with two convolutional blocks and both the spatial attention and subject layers of Défossez et al. (2022), affine projection temporal aggregation layer with a single block in the CLIP projection head, and adapted CLIP loss from Défossez et al. (2022) *i.e.*, with normalization along the image axis only, the brain-to-image term only (first term of Eq. 1) and a fixed temperature parameter $\tau = 1$. The final architecture configuration is presented in Table S1.

## A.3 IMAGE EMBEDDINGS

We evaluate the performance of linear baselines and of a deep convolutional neural network on the MEG-to-image retrieval task using a set of classic visual embeddings. We grouped these embeddings by their corresponding paradigm:

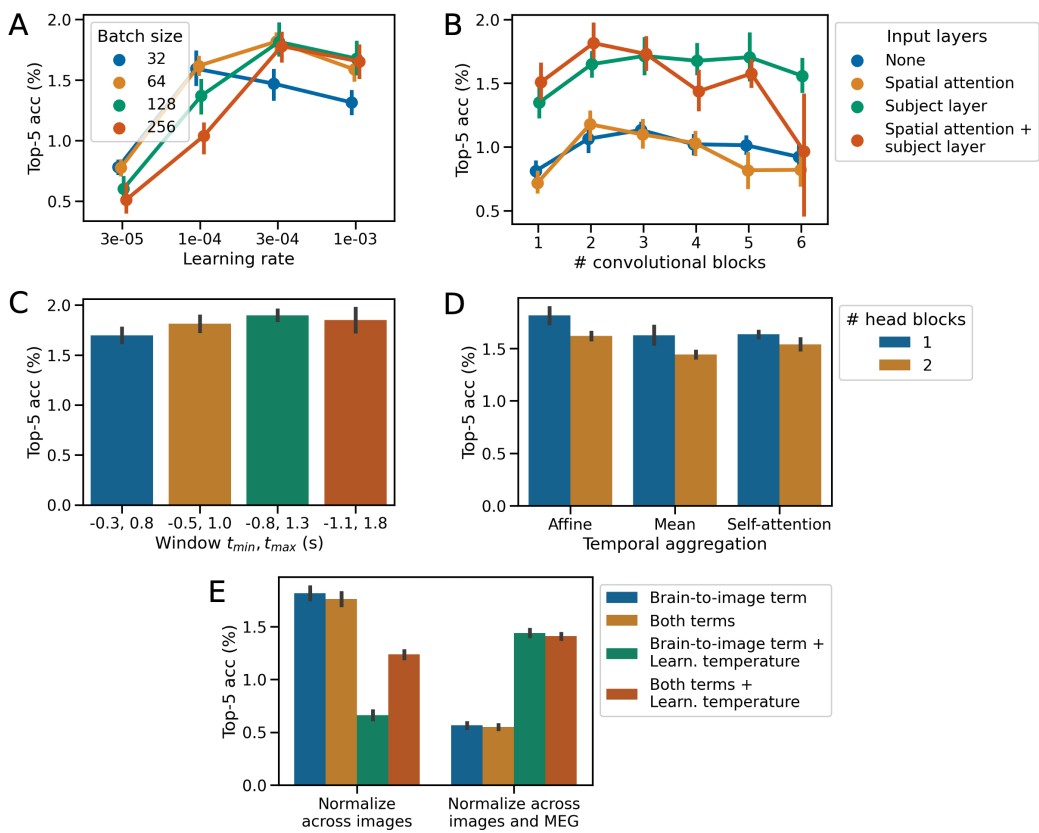

Figure S1: Hyperparameter search results for the MEG-to-image retrieval task, presenting the impact of (**A**) optimizer learning rate and batch size, (**B**) number of convolutional blocks and use of spatial attention and/or subject-specific layers in the brain module, (**C**) MEG window parameters, (**D**) type of temporal aggregation layer and number of blocks in the CLIP projection head of the brain module, and (**E**) CLIP loss configuration (normalization axes, use of learned temperature parameter and use of symmetric terms). Chance-level performance top-5 accuracy is 0.05%.

Table S1: Brain module configuration adapted from Défossez et al. (2022) for use with a target latent of size 768 (*e.g.* CLIP-Vision (CLS), see Section 2.4) in retrieval settings.

| Layer | Input shape | Output shape | # parameters |
|---|---|---|---|
| Spatial attention block | (272, 181) | (270, 181) | 552,960 |
| Linear projection | (270, 181) | (270, 181) | 73,170 |
| Subject-specific linear layer | (270, 181) | (270, 181) | 291,600 |
| Residual dilated conv block 1 | (270, 181) | (320, 181) | 1,183,360 |
| Residual dilated conv block 2 | (320, 181) | (320, 181) | 1,231,360 |
| Linear projection | (320, 181) | (2048, 181) | 1,518,208 |
| Temporal aggregation | (2048, 181) | (2048, 1) | 182 |
| MLP projector | (2048, 1) | (768, 1) | 1,573,632 |
| Total | | | 6,424,472 |

**Supervised learning.** The last layer, with dimension 1000, of VGG-19.

**Text/Image alignment.** The last hidden layer of CLIP-Vision (257x768), CLIP-Text (77x768), and their CLS and MEAN pooling.

**Self-supervised learning.** The output layers of DINOv1, DINOv2 and their CLS and MEAN pooling. The best-performing DINOv2 variation reported in tables and figures is ViT-g/14.

**Variational autoencoders.** The activations of the 31 first layers of the very deep variational-autoencoder (VDVAE), and the bottleneck layer (4x64x64) of the Kullback-Leibler variational-autoencoder (AutoKL) used in the generative module (Section 2.5).

**Engineered features.** The color histogram of the seen image (8 bins per channels); the local binary patterns (LBP) using the implementation in OpenCV 2 (Bradski, 2000) with 'uniform' method, $P = 8$ and $R = 1$; the Histogram of Oriented Gradients (HOG) using the implementation of sk-image (Van der Walt et al., 2014) with 8 orientations, 8 pixels-per-cell and 2 cells-per-block.

## A.4  7T FMRI DATASET

The Natural Scenes Dataset (NSD) (Allen et al., 2022) contains fMRI data from 8 participants viewing a total of 73,000 RGB images. It has been successfully used for reconstructing seen images from fMRI in several studies (Takagi & Nishimoto, 2023; Ozcelik & VanRullen, 2023; Scotti et al., 2023). In particular, these studies use a highly preprocessed, compact version of fMRI data ("betas") obtained through generalized linear models fitted across multiple repetitions of the same image.

Each participant saw a total of 10,000 unique images (repeated 3 times each) across 37 sessions. Each session consisted in 12 runs of 5 minutes each, where each image was seen during 3 s, with a 1-s blank interval between two successive image presentations. Among the 8 participants, only 4 (namely 1, 2, 5 and 7) completed all sessions.

To compute the three latents used to reconstruct the seen images from fMRI data (as described in Section 2.5) we follow Ozcelik & VanRullen (2023) and train and evaluate three distinct Ridge regression models using the exact same split. That is, for each of the four remaining participants, the 9,000 uniquely-seen-per-participant images (and their three repetitions) are used for training, and a common set of 1000 images seen by all participant is kept for evaluation (also with their three repetitions). We report reconstructions and metrics for participant 1.

The $\alpha$ coefficient for the $L2$-regularization of the regressions are cross-validated with a 5-fold scheme on the training set of each subject. We follow the same standardization scheme for inputs and predictions as in Ozcelik & VanRullen (2023).

Fig. S2 presents generated images obtained using the NSD dataset (Allen et al., 2022).

## A.5  LINEAR RIDGE REGRESSION SCORES ON PRETRAINED IMAGE REPRESENTATIONS

We provide a (5-fold cross-validated) Ridge regression baseline (Table S2) for comparison with our brain module results of Section 3, showing considerable improvements for the latter.

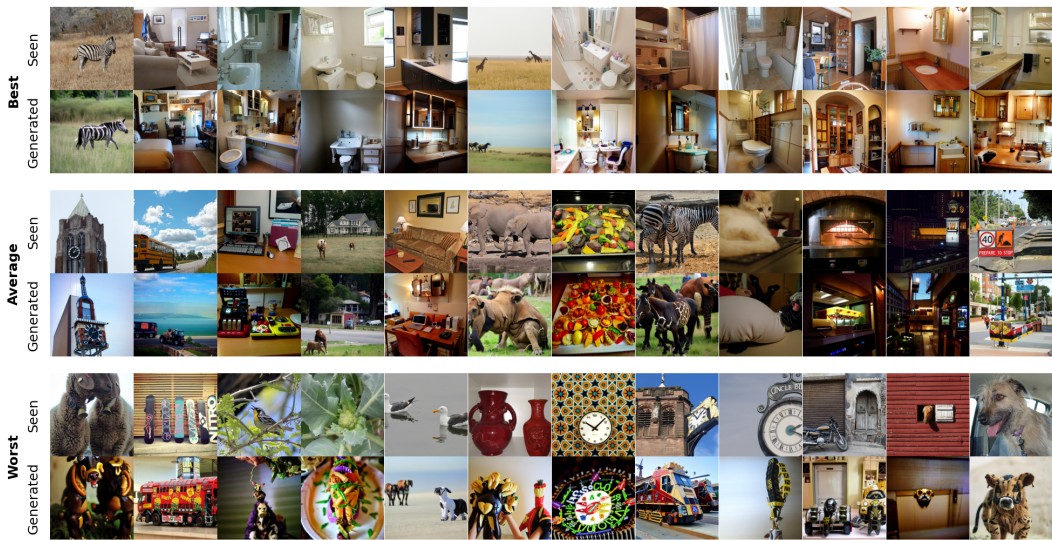

Figure S2: Examples of generated images conditioned on fMRI-based latent predictions. The groups of three stacked rows represent best, average and worst retrievals, as evaluated by the sum of (minus) SwAV and SSIM.

Table S2: Image retrieval performance of a linear Ridge regression baseline on pretrained image representations.

| Latent kind | Latent name | Top-5 acc (%) ↑ | | Median relative rank ↓ | |
|---|---|---|---|---|---|
| | | Small set | Large set | Small set | Large set |
| Text/Image alignment | CLIP-Vision (CLS) | 10.5 | 0.50 | 0.23 | 0.34 |
| | CLIP-Text (mean) | 6.0 | 0.25 | 0.42 | 0.43 |
| | CLIP-Vision (mean) | 5.5 | 0.46 | 0.32 | 0.37 |
| Feature engineering | Color histogram | 7.0 | 0.33 | 0.31 | 0.40 |
| | Local binary patterns (LBP) | 3.5 | 0.37 | 0.34 | 0.44 |
| | FFT 2D (as real) | 4.5 | 0.46 | 0.40 | 0.45 |
| | HOG | 3.0 | 0.42 | 0.45 | 0.46 |
| | FFT 2D (log-PSD and angle) | 2.0 | 0.37 | 0.47 | 0.46 |
| Variational autoencoder | AutoKL | 7.5 | 0.54 | 0.24 | 0.38 |
| | VDVAE | 8.0 | 0.50 | 0.33 | 0.43 |
| Self-supervised learning | DINOv2 (CLS) | 7.5 | 0.46 | 0.25 | 0.35 |
| Supervised | VGG-19 | 11.5 | 0.67 | 0.17 | 0.31 |

### A.6 IMPACT OF CHOICE OF LAYER IN SUPERVISED MODELS

We replicate the analysis of Fig. 2 on different layers of the supervised model (VGG-19). As shown in Table S3, some of these layers slightly outperform the last layer. Future work remains necessary to further probe which layer, or which combination of layers and models may be optimal to retrieve images from brain activity.

### A.7 MEG-BASED IMAGE RETRIEVAL EXAMPLES

Fig. S3 shows examples of retrieved images based on the best performing latents identified in Section 3.

Table S3: Image retrieval performance of intermediate image representations of the VGG-19 supervised model.

| Latent kind | Latent name | Top-5 acc (%) ↑ | | Median relative rank ↓ | |
|---|---|---|---|---|---|
| | | Small set | Large set | Small set | Large set |
| | VGG-19 (last layer) | 70.333 | 12.292 | 0.005 | 0.013 |
| | VGG-19 (avgpool) | 73.833 | 17.417 | 0.000 | 0.006 |
| Supervised | VGG-19 (classifier_dropout_2) | 73.833 | 17.375 | 0.000 | 0.005 |
| | VGG-19 (classifier_dropout_5) | 74.500 | 16.403 | 0.000 | 0.007 |
| | VGG-19 (maxpool2d_35) | 64.333 | 13.278 | 0.005 | 0.014 |

To get a better sense of what time-resolved retrieval yields in practice, we present the top-1 retrieved images from an augmented retrieval set built by concatenating the "large" test set with an additional set of 3,659 images that were not seen by the participants (Fig. S4).

### A.8 MEG-BASED IMAGE GENERATION EXAMPLES

Fig. S5 shows representative examples of generated images obtained with our diffusion pipeline[3].

Fig. S6 specifically shows examples of failed generations. Overall, they appear to encompass different types of failures. Some generations appear to miss the correct category of the true object (*e.g.* bamboo, batteries, bullets and extinguisher in columns 1-4), but generate images with partially similar textures. Other generations appear to recover some category-level features but generate unrealistic chimeras (bed: weird furniture, alligator: swamp beast; etc. in columns 5-6). Finally, some generations seem to be completely wrong, with little-to-no preservation of low- or high-level features (columns 7-8). We speculate that these different types of failures may be partially resolved with different methods, such as better generation modules (for chimeras) and optimization on both low- and high-level features (for category errors).

### A.9 PERFORMANCE OF TEMPORALLY-RESOLVED IMAGE RETRIEVAL WITH GROWING WINDOWS

To complement the results of Fig. 3 on temporally-resolved retrieval with sliding windows, we provide a similar analysis in Fig. S7, instead using growing windows. Beginning with the window spanning -100 to 0 ms around image onset, we grow it by increments of 25 ms until it spans both stimulus presentation and interstimulus interval regions (*i.e.*, -100 to 1,500 ms). Separate models are finally trained on each resulting window configuration.

Consistent with the decoding peaks observed after image onset and offset (Fig. 3), the retrieval performance of all growing-window models considerably improves after the offset of the image. Together, these results suggest that the brain activity represents both low- and high-level features even after image offset. This finding clarifies mixed results previously reported in the literature. Carlson et al. (2011; 2013) reported small but significant decoding performances after image offset. However, other studies (Cichy et al., 2014; Hebart et al., 2023) did not observe such a phenomenon. In all these cases, decoders were based on pairwise classification of object categories and on linear classifiers. The improved sensitivity brought by (1) our deep learning architecture, (2) its retrieval objective and (3) its use of pretrained latent features may thus help clarify the dynamics of visual representations in particular at image offset. We speculate that such offset responses could reflect an intricate interplay between low- and high-level processes that may be difficult to detect with a pairwise linear classifier. We hope that the present methodological contribution will help shine light on this understudied phenomenon.

---

[3]Images may look slightly different from those in Fig. 4 due to different random seeding.

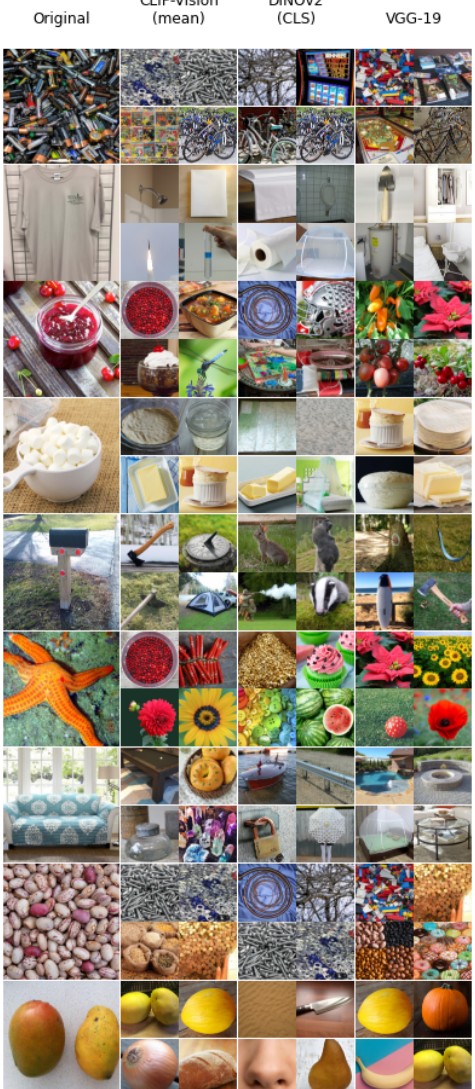

Figure S3: Representative examples of retrievals (top-4) using models trained on full windows (from -0.5 s to 1 s after image onset). Retrieval set: $N =$6,059 images from 1,196 categories.

## A.10    PER-PARTICIPANT IMAGE GENERATION PERFORMANCE

Table S4 provides the image generation metrics at participant-level. For each participant, we compute metrics over the 200 generated images obtained by averaging the outputs of the brain module for all 12 presentations of the stimulus.

## A.11    ANALYSIS OF TEMPORAL AGGREGATION LAYER WEIGHTS

We inspect our decoders to better understand how they use information in the time domain. To do so, we leverage the fact that our architecture preserves the temporal dimension of the input up until the output of its convolutional blocks. This output is then reduced by an affine transformation learned by the temporal aggregation layer (see Section 2.3 and Appendix A.1). Consequently, the weights $\boldsymbol{w}^{agg} \in \mathbb{R}^T$ can reveal on which time steps the models learned to focus. To facilitate inspection, we

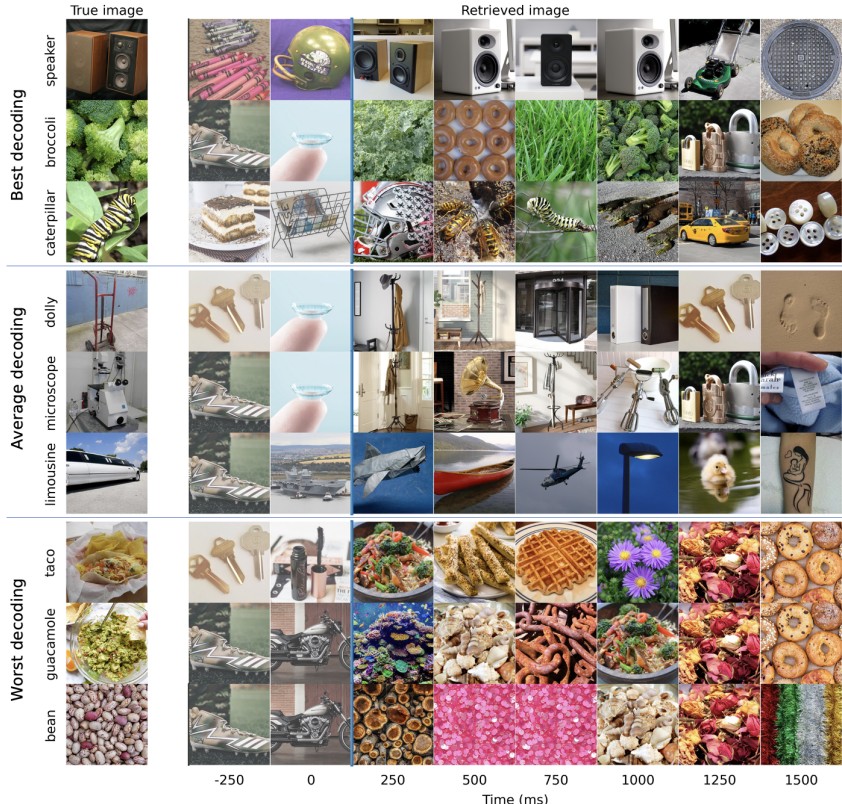

Figure S4: Representative examples of dynamic retrievals using CLIP-Vision (CLS) and models trained on 250-ms non-overlapping sliding windows (Image onset: $t = 0$, retrieval set: $N =6,059$ from 1,196 categories). The groups of three stacked rows represent best, average and worst retrievals, obtained by sampling examples from the $<10\%$, 45-55% and $>90\%$ percentile groups based on top-5 accuracy.

Table S4: Quantitative evaluation of reconstruction quality from MEG data on THINGS-MEG for each participant. We use the same metrics as in Table 1.

| Participant | Low-level | | | High-level | | | |
|---|---|---|---|---|---|---|---|
| | PixCorr ↑ | SSIM ↑ | AlexNet(2) ↑ | AlexNet(5) ↑ | Inception ↑ | CLIP ↑ | SwAV ↓ |
| 1 | 0.070 ± 0.009 | 0.338 ± 0.015 | 0.741 | 0.814 | 0.672 | 0.768 | 0.590 ± 0.007 |
| 2 | 0.081 ± 0.010 | 0.341 ± 0.015 | 0.788 | 0.879 | 0.710 | 0.799 | 0.560 ± 0.008 |
| 3 | 0.073 ± 0.010 | 0.335 ± 0.015 | 0.725 | 0.825 | 0.675 | 0.770 | 0.588 ± 0.008 |
| 4 | 0.082 ± 0.009 | 0.328 ± 0.014 | 0.701 | 0.797 | 0.634 | 0.744 | 0.599 ± 0.008 |

initialize $\boldsymbol{w}^{agg}$ to zeros before training and plot the mean absolute weights of each model (averaged across seeds).

The results are presented in Fig. S8. While these weights are close to zero before stimulus onset, they deviate from this baseline after stimulus onset, during the maintenance period and after stimulus offset. Interestingly, and unlike high-level features (*e.g.* VGG-19, CLIP-Vision), low-level features (*e.g.* color histogram, AutoKL and DINOv2) have close-to-zero weights in the 0.2-0.5 s interval.

This result suggests that low-level representations quickly fade away at that moment. Overall, this analysis demonstrates that the models rely on these three time periods to maximize decoding performance, including the early low-level responses ($t =$0-0.1 s).

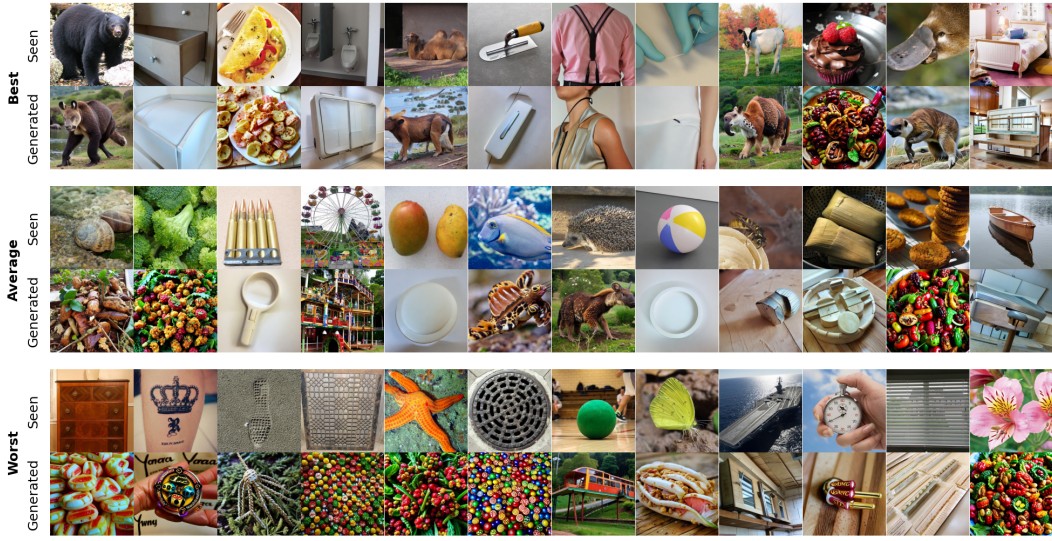

Figure S5: Representative examples of generated images conditioned on MEG-based latent predictions. The groups of three stacked rows represent best, average and worst generations, as evaluated by the sum of (minus) SwAV and SSIM.

## A.12 TEMPORALLY-RESOLVED IMAGE GENERATION METRICS

Akin to the time-resolved analysis of retrieval performance shown in Fig. 3, we evaluate the image reconstruction metrics used in Table 1 on models trained on 100-ms sliding windows. Results are shown in Fig. S9.

Low-level metrics peak in the first 200 ms while high-level metrics reach a performance plateau that is maintained throughout the image presentation interval. As seen in previous analyses (Fig. 3, S7 and S8), a sharp performance peak is visible for low-level metrics after image offset.

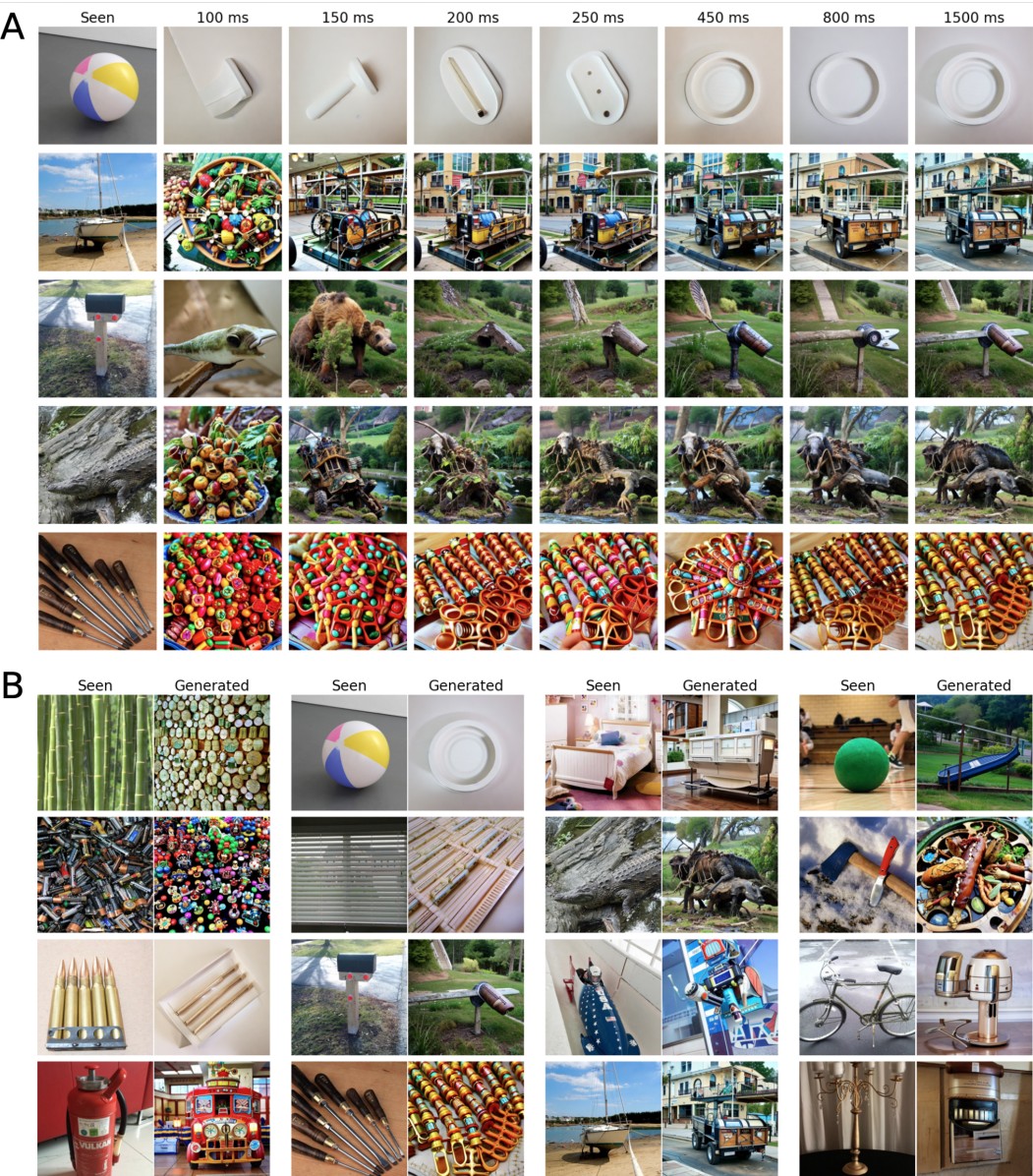

Figure S6: Examples of failed generations. (**A**) Generations obtained on growing windows starting at image onset (0 ms) and ending at the specified time. (**B**) Full-window generations (-500 to 1,000 ms).

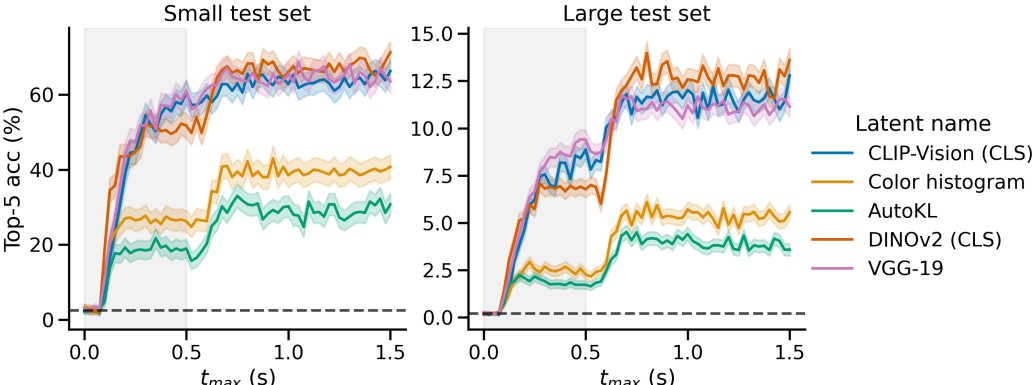

Figure S7: Retrieval performance of models trained on growing windows (from -100 ms up to 1,500 ms relative to stimulus onset) for different image embeddings. The shaded gray area indicates the 500-ms interval during which images were presented to the participants and the horizontal dashed line indicates chance-level performance. Accuracy plateaus a few hundreds of milliseconds after both image onset and offset.

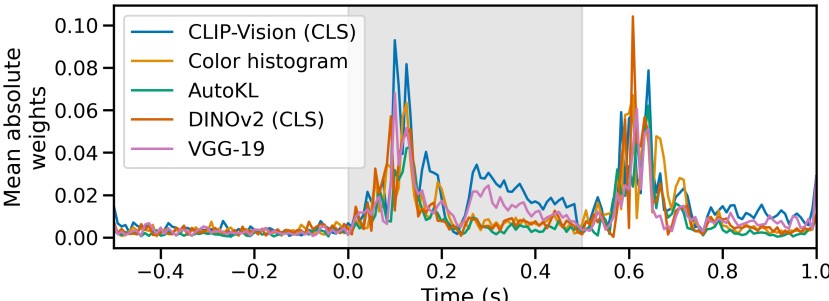

Figure S8: Mean absolute weights learned by the temporal aggregation layer of the brain module. Retrieval models were trained on five different latents. The absolute value of the weights of the affine transformation learned by the temporal aggregation layer were then averaged across random seeds and plotted against the corresponding timesteps. The shaded gray area indicates the 500-ms interval during which images were presented to the participants.

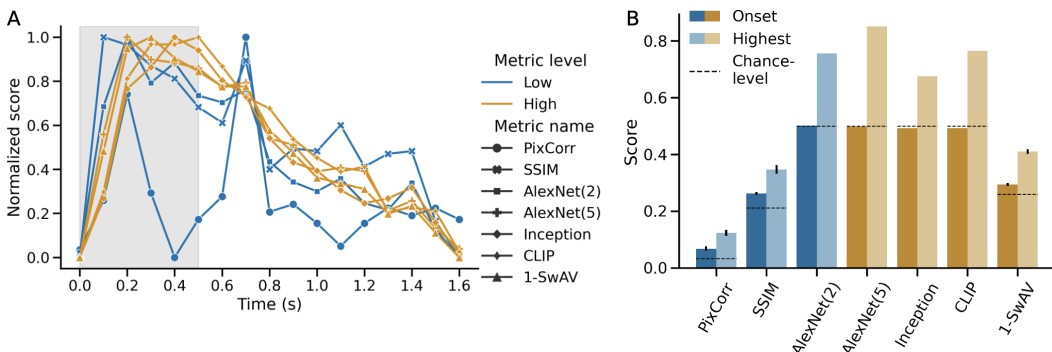

Figure S9: Temporally-resolved evaluation of reconstruction quality from MEG data. We use the same metrics as in Table 1 to evaluate generation performance from sliding windows of 100 ms with no overlap. (**A**) Normalized metric scores (min-max scaling between 0 and 1, metric-wise) across the post-stimulus interval. (**B**) Unnormalized scores comparing, for each metric, the score at stimulus onset and the maximum score obtained across all windows in the post-stimulus interval. Dashed lines indicate chance-level performance and error bars indicate the standard error of the mean for PixCorr, SSIM and SwAV.