# OpenReview forum: "Brain decoding: toward real-time reconstruction of visual perception"
_ICLR.cc/2024/Conference — ICLR 2024 poster_

### Official Review · Reviewer_F12m · 2023-10-31

**Soundness:** 3 good
**Presentation:** 3 good
**Contribution:** 2 fair
**Rating:** 6
**Confidence:** 3

**Summary:**

The authors perform MEG conditioned visual decoding.

Compared to other works that leverage fMRI, MEG is a different information source that presents unique challenges.

The authors use an align then generate strategy, where they learn a function that takes as input the MEG signal, and train it to align with a CLIP latent using a weighted sum of infoNCE and MSE loss. For image generation, they use Versatile Diffusion and regress the needed conditioning variables from MEG.

They observe that it is possible to recover high level semantics in the reconstructed images.

**Strengths:**

MEG decoding of full images is an under-explored area compared to fMRI based decoding, and it is a harder task, given the low channel count relative to the tens of thousands of voxels in fMRI. To my knowledge, this is the first time that image decoding has been demonstrated using MEG.

The paper is methodologically sound, outlining different training objectives for different parts of the proposed pipeline. The paper provides systematic benchmarks, showing that their MEG decoder leads to reasonable image retrieval and image generation.

I applaud the authors for showing "representative" retrieval and best/mean/worst decoding results, which helps gauge the effectiveness of the method.

It is also interesting that they found MEG capable of recovering high level semantics. Although it is not fully clear if this is a limitation of MEG or their method (should probably discuss more).

**Weaknesses:**

As with other deep decoding papers, it is not super clear what the ultimate scientific insight is. This is not a criticism specific to this paper, but more generally aimed at current decoding works which leverage powerful image priors and deep non-linear decoding/embedding functions.

In this aspect, this paper is better than most, as their Figure 3 provides some insight on the temporal dynamics of decoding. I think it would benefit the paper to add some discussion (not necessarily experiments) on extending this to EEG based decoding, or other potential practical applications or scientific insights.

**General clarifications:**

The clarity of many of their methods could be improved. The author repeatedly references Defossez et al. in reference to their methods. But in the main text it is not super clear. Concretely I would like the authors to clarify the following:
1. What the the MEG conv "encoder" convolving over?

2. How do you combine the MEG channels?

3. What temporal aggregation layer did you use? You mention global pooling, affine, and attention. Which layer did you end up using? Because you discuss this and then never talk about which method you ended up using.

4. How do the different aggregation layers work? I ask this question in the context of Figure 3. Because you discuss using a 1500ms window, then shift to a 250ms window. Do you train a new model? Do you re-use the 1500ms model but change the aggregation? If you do train a new model, are you taking multiple 250ms windows and supervising with the same image target? For the sliding window, what is the step size?

5. For Figure 2, in the supervised models (VGG, ResNet, etc.) are you using the last layer (1000 imagenet classes layer), or the post-pooled layer.

6. For retrieval, are you always using cosine/dot-product similarity?

**Minor format error:**
1. The authors have ICLR 2023 in the header, when it should be ICLR 2024. And they have line numbers, which do not seem to be present in the default ICLR template.

**Minor clarifications:**
1. Can you clarify if $N$ (line 75) denotes the number of images? It doesn't seem like you define $N$ prior/after using it.
2. To provide more context, can you mention in line 84 that you are using the infoNCE loss, rather than just mentioning the CLIP loss.
3. In section 2.2, can you clarify if you are normalizing $\hat{z}$ to norm = 1 for eq. 1, and not assuming a fixed norm for eq. 2? Otherwise it seems like the two losses would have trivially the same optima, but I guess you are trying to have one loss align the direction, and have a second loss align the direction + norm.

**Additional citations:**

The author discusses one approach towards decoding, but I would appreciate if the author could also discuss the brain gradient conditioned image generation work listed below, the most recent of which also leverage GANs/Diffusion models:

Inception loops discover what excites neurons most using deep predictive models (**Nature 2019**); Neural population control via deep image synthesis (**Science 2019**); Evolving Images for Visual Neurons Using a Deep Generative Network Reveals Coding Principles and Neuronal Preferences (**Cell 2019**); Computational models of category-selective brain regions enable high-throughput tests of selectivity (**Nature Communications 2021**); NeuroGen: Activation optimized image synthesis for discovery neuroscience (**Neuroimage 2022**); Brain Diffusion for Visual Exploration: Cortical Discovery using Large Scale Generative Models (**NeurIPS 2023**); Energy Guided Diffusion for Generating Neurally Exciting Images (**NeurIPS 2023**)

Overall I think the paper is sound, interesting, and provides good insight on neural decoding from an often overlooked modality.

**Questions:**

Please see the weakness's section for questions.

---

> ### Author Response · Authors · 2023-11-21
>
> We thank the reviewer for the thorough review and insightful suggestions that effectively help improve the quality of our manuscript.
>
> # 1. Weakness
>
> We agree with the reviewer that the scientific insight provided by our study could have been clearer. We have now (1) updated the contribution section, (2) provided new analyses and (3) amended the discussion section.
>
> ## 1.A. Contribution
>
> This study primarily focuses on methodological contributions. We have now amended the manuscript as follows:
>
> > “Our approach provides three main contributions: our MEG decoder (1) yields a 7X increase in performance as compared to linear baselines (Figure 2), (2) helps reveal when high-level semantic features are processed in the brain (Figure 3) and (3) allows the continuous generation of images from temporally-resolved brain signals (Figure 4). Overall, this approach thus paves the way to better understand the unfolding of the brain responses to visual inputs.“

---

> ### Author Response · Authors · 2023-11-21
>
> ## 1.B. New analyses
>
> To highlight the scientific insights that our MEG decoding may bring, we implemented several new analyses and updated:
> Figure 3, which now shows retrieval performance at a much higher temporal resolution;
> Figure 4, which now shows image generations as a function of time;
> Supplementary figures (growing window retrieval, model weights analysis and time-resolved image generation metrics in Appendices 8,10 and 11, respectively), which detail when low- and high-level features may be decoded from brain activity.
>
> Overall, these new analyses reveal that (1) late brain responses specifically represent high-level features and (2) the brain responses to image offset represent both low and high-level responses. These elements subsume previous neuroscientific investigation. We now indicate:
>
> In Appendix 5:
>
> > “Consistent with the decoding peaks observed after image onset and offset (Fig. 3), the retrieval performance of all growing-window models considerably improves after the offset of the image. Together, these results suggest that the brain activity represents both low- and high-level features even after image offset. This finding clarifies mixed results previously reported in the literature. Carlson et al., (2011; 2013), reported small but significant decoding performances after image offset. However, other studies (Cichy et al., 2014, Hebart et al., 2023) did not observe such a phenomenon. In all these cases, decoders were based on pairwise classification of object categories and on linear classifiers. The improved sensitivity brought by (1) our deep learning architecture, (2) its retrieval objective and (3) its use of pretrained latent features may thus help clarify the dynamics of visual representations in particular at image offset. We speculate that such offset responses could reflect an intricate interplay between low- and high-level processes that may be difficult to detect with a pairwise linear classifier. We hope that the present methodological contribution will help shine light on this understudied phenomenon.”
>
> References:
> ```Carlson, Thomas A., et al. "High temporal resolution decoding of object position and category." Journal of vision 11.10 (2011): 9-9.
> Carlson, Thomas, et al. "Representational dynamics of object vision: the first 1000 ms." Journal of vision 13.10 (2013): 1-1.
> Cichy, Radoslaw Martin, Dimitrios Pantazis, and Aude Oliva. "Resolving human object recognition in space and time." Nature neuroscience 17.3 (2014): 455-462.
> Hebart, Martin N., et al. "THINGS-data, a multimodal collection of large-scale datasets for investigating object representations in human brain and behavior." Elife 12 (2023): e82580.
> ```
>
> In Appendix 10:
>
> > “We inspect our decoders to better understand how they use information in the time domain. To do so, we leverage the fact that our architecture preserves the temporal dimension of the input up until the output of its convolutional blocks. This output is then reduced by an affine transformation learned by the temporal aggregation layer (see Section 2.3 and Appendix A.2). Consequently, the weights $w^{agg} \in \mathbb{R}^T$ can reveal on which time steps the models learned to focus. To facilitate inspection, we initialize $w^{agg}$ to zeros before training and plot the mean absolute weights of each model (averaged across seeds).
>
> > The results are presented in Fig. S8. While these weights are close to zero before stimulus onset, they deviate from this baseline after stimulus onset, during the maintenance period and after stimulus offset. Interestingly, and unlike high-level features (e.g., VGG-19, CLIP-Vision), low-level features (e.g., color histogram, AutoKL and DINOv2) have close-to-zero weights in the 0.2-0.5 s interval.
>
> > This result suggests that low-level representations quickly fade away at that moment. Overall, this analysis demonstrates that the models rely on these three time periods to maximize decoding performance, including the early low-level responses ($t=0-0.1$s).“
>
> Finally, we now provide a time-resolved analysis of low- and high-level reconstruction metrics (Appendix A.11). These results confirm that the early responses to both image onset and offset are primarily associated with low-level metrics. On the other hand, the decoding of brain responses during the second half of the image presentation interval (0.2-0.5 s) appears to be more related to high-level features.

---

> ### Author Response · Authors · 2023-11-21
>
> ## 1.C. Updated impact
>
> Finally, we modified the manuscript to further discuss what scientific insights our approach may provide. The present work is primarily a methodological contribution, and a demonstration that natural images can be decoded from MEG at a much better level than anticipated (7X as compared to linear baseline). We updated the discussion section (paragraph on Impact):
>
> > “Our methodological contribution has both fundamental and practical impacts. First, decoding the unfolding of perceptual representations could clarify the unfolding of visual processing in the brain. While there is considerable work on this issue, neural representations are challenging to interpret because they represent latent, abstract, feature spaces. Generative decoding, on the contrary, can provide concrete and, thus, interpretable predictions. Put simply, generating images at each time step could help neuroscientists understand whether specific – potentially unanticipated – textures or object parts are represented. For example, Cheng et al., (2023) showed that generative decoding applied to fMRI can be used to decode the subjective perception of visual illusions. Such techniques can thus help to clarify the neural bases of subjective perception and to dissociate them from those responsible for ``copying’’ sensory inputs. Our work shows that this endeavor could now be applied to clarify *when* these subjective representations arise.
>
> > Second, generative brain decoding has concrete applications. For example, it has been used in conjunction with encoding, to identify stimuli that maximize brain activity (Bashivan et al., 2019). Furthermore, non-invasive brain-computer interfaces (BCI) have been long-awaited by patients with communication challenges related to brain lesions. BCI, however, requires real-time decoding, and thus limits the use of neuroimaging modalities with low temporal resolution such as fMRI. This application direction, however, will likely require extending our work to EEG, which provides similar temporal resolution to MEG, but is typically much more common in clinical settings.“
>
> ```
> References
> Cheng, Fan L., et al. "Reconstructing visual illusory experiences from human brain activity." Science Advances 9.46 (2023): eadj3906.
> ```

---

> > ### Author Response · Authors · 2023-11-21
> >
> > # 2. General clarifications
> > ## 2.1 Description of the brain module
> > We thank the reviewer for pointing out this opportunity to improve the clarity of the manuscript. We added a more detailed description of (1) the brain module, (2) the different temporal aggregation approaches considered and (3) the way different windows are handled to the manuscript. We also answer the reviewer’s questions directly below.
> >
> > In Appendix A.1:
> >
> > > “The brain module first applies two successive linear transformations in the spatial dimension to an input MEG window. The first linear transformation is the output of an attention layer conditioned on the MEG sensor positions. The second linear transformation is learned subject-wise, such that each subject ends up with their own linear projection matrix $W_s \in \mathbb{R}^{C \times C}$, with $C$ the number of input MEG channels and $s \in \[1, S\]$ where $S$ is the number of subjects. The module then applies a succession of 1D convolutional blocks that operate in the temporal dimension and treat the spatial dimension as features. These blocks each contain three convolutional layers (dilated kernel size of 3, stride of 1) with residual skip connections. The first two layers of each block use GELU activations while the last one use a GLU activation. The output of the last convolutional block is passed through a learned linear projection to yield a different number of features $F'$ (fixed to 2048 in our experiments).
> >
> > > The resulting features are then fed to a temporal aggregation layer which reduces the remaining temporal dimension. Given the output of the brain module backbone $Y_{backbone} \in \mathbb{R}^{F' \times T}$, we compare three approaches to reduce the temporal dimension of size $T$:
> > (1) Global average pooling, ie the features are averaged across time steps;
> > (2) Learned affine projection in which the temporal dimension is projected from $ \mathbb{R}^T$ to $ \mathbb{R}$ using a learned weight vector $w^{agg} \in  \mathbb{R}^{T}$ and bias $b^{agg} \in  \mathbb{R}$;
> > (3) Bahdanau attention layer (Bahdanau et al., 2014) which predicts an affine projection from $ \mathbb{R}^T$ to $ \mathbb{R}$ conditioned on the input  $Y_{backbone}$ itself.
> > Following the hyperparameter search of Appendix A.2, we selected the learned affine projection approach for our experiments.
> > Finally, the resulting output is fed to CLIP and MSE head-specific MLP projection heads where a head consists of repeated LayerNorm-GELU-Linear blocks, to project from $F'$ to the $F$ dimensions of the target latent.
> >
> > > We refer the interested reader to Défossez et al., (2023) for a description of the original architecture, and to the code available at [http://github.com/facebookresearch/brainmagick](http://github.com/facebookresearch/brainmagick]).”
> >
> > References:
> > ```
> > Bahdanau, Dzmitry, Kyunghyun Cho, and Yoshua Bengio. "Neural machine translation by jointly learning to align and translate."  arXiv:1409.0473 (2014).
> > ```
> >
> > We further answer the specific questions below.
> >
> > 1. The brain module contains the equivalent of convolutions over both the spatial and temporal dimensions. Specifically, its convolutional blocks operate in the temporal dimension and treat the spatial dimension as features.
> >
> > 2. The MEG channels are first combined using the projection matrix output by the spatial attention layer. They are then mixed again using a subject-specific linear transformation. The following convolutional blocks then mix information in both the temporal and feature (which correspond to the spatial dimension for the first convolutional block) dimensions.
> >
> > 3. Following the hyperparameter search presented in Appendix A.2, we used the affine projection temporal aggregation layer. We included further information in the detailed description of the brain module now in Appendix A.1 and pasted above.
> >
> > 4. The different aggregation layer approaches are now described in Appendix A.1 (cf above). We indeed train separate models on different window sizes. In practice, this means that we trained multiple versions of each model to accommodate different window sizes, e.g., 1.5 s for Fig. 2, and 100 ms with stride of 25 ms for Fig. 3 (please note that this used to be non-overlapping 250-ms windows in the original submission but has been updated as mentioned in our answer to point 1 above). Despite receiving a different window as input, these models use the exact same targets, i.e. the latent representation of the corresponding image.
> >
> > Sharing a single model across window sizes is very interesting and should indeed be explored in future work. Overall, we added the following description to Section 2.3 to clarify this important point:
> >
> > > “Of note, when comparing performance on different window configurations \eg to study the dynamics of visual processing in the brain, we train a different model per window configuration. Despite receiving a different window of MEG as input, these models use the same latent representations of the corresponding images.”

---

> > > ### Author Response · Authors · 2023-11-21
> > >
> > > ## 2.2 Supervised image models
> > > For each embedding used in the paper, we now added a reference to the layer of the corresponding deep net in Appendix A.3. In particular, for all supervised learning models, this is the last layer, of dimension 1000.
> > >
> > > ## 2.3. Metric used for retrieval
> > >
> > > When training models with the CLIP loss, we implicitly used the cosine similarity as feature vectors are first standardized before computing their dot product (Radford et al., 2021). Similarly, for evaluation, we also relied on the cosine similarity to compute retrieval metrics (relative median rank and top-5 accuracy). We updated the text in Section 2.7 accordingly:
> > >
> > > > “We evaluate retrieval using either the relative median rank (which does not depend on the size of the retrieval set), defined as the rank of a prediction divided by the size of the retrieval set, or the top-5 accuracy (which is more common in the literature). In both cases, we use cosine similarity to evaluate the strength of similarity between feature representations (Radford et al., 2021).”
> > >
> > > ## 2.4. Formatting errors
> > > Thank you for pointing out these errors. We have fixed them in the new version of the manuscript.
> > >
> > > ## 2.5 Clarifications
> > > - Indeed, $N$ refers to the number of images. We updated the text accordingly.
> > > - We added a mention and reference to the infoNCE loss: “To do so, we train $\textbf{f}_\theta$ using the CLIP loss (Radford et al., 2021) (i.e., the InfoNCE loss (Oord et al., 2018) applied in both brain-to-image and image-to-brain directions) on batches of size B with exactly one positive example [...]”
> > > - We thank the reviewer for pointing this out. We updated the text accordingly: “This is done using a standard mean squared error (MSE) loss over the (unnormalized) $z_i$ and $z_i$(hat): [...]”
> > >
> > > ## 2.6 Additional references
> > > We thank the reviewer for providing these references. The objective of these studies, however, is substantially different from our contribution, as it is not specific to the time domain, and requires a combination of encoding and decoding tools. We thus simply added the following element to the Impact section of the discussion:
> > >
> > > > “For example, it has been used in conjunction with encoding, to identify stimuli that maximize brain activity (Bashivan et al., 2019).”

---

> ### Comment · Reviewer_F12m · 2023-11-21
> **Thank you**
>
> I've increased my score to 8, I am more convinced of the scientific contribution.
>
> I still want to comment on the sub-optimal presentation of the rebuttal itself.
>
> While the authors have indeed provided additional scientific justification which I find convincing, the authors have not addressed my requests for clarification, and the rebuttal is not well formatted.
>
> The revision PDF also does not include highlights as typical in ICLR rebuttals to visualize the changes.
>
> Hopefully the authors can improve upon this in the future.
>
> Edit: Reading the additional clarifications below, the authors use the last layer (1000 class output) of the each network as the baseline, which does not strike me as particularly sound. In this layer, there exists an exact optima, where it is an indicator function of the correct class. It is not clear that this output is necessarily reflective of different architectures. And to my knowledge, using a linear probe on the last categorical layer is not done in other works. I'm re-calibrating my score to a 6.

---

> > ### Author Response · Authors · 2023-11-21
> >
> > Thank you for your prompt feedback. It seems like there might have been a delay between the submission of the first part of our answers and the remaining parts. We hope you will find the comments that were since posted helpful in clarifying our work.
> >
> > We are currently working on providing a diff version of the pdf and should be able to post it soon.
> >
> > Finally, we will work on improving the formatting of the rebuttal following your comment.

---

> > > ### Comment · Reviewer_F12m · 2023-11-21
> > >
> > > Ah that makes sense. I was looking at the rebuttal and only the first half showed up.
> > >
> > > Thanks for clarifying the contributions.

---

> > ### Author Response · Authors · 2023-11-22
> >
> > Thank you for raising this new concern. We clarify below that this issue:
> > 1. Is orthogonal to our main objective, namely generating images from MEG recordings;
> > 2. Does not affect the validity of the retrieval analysis (Figure 2), and only concerns a small subset of the supervised models (e.g. VGG, ResNet).
> >
> > However, we do agree that it is interesting to further explore the impact of layer selection for supervised classifiers.
> >
> > Consequently, we
> > - will update the manuscript to clarify this important research direction;
> > - replicated these retrieval analyses for 4 hidden layers of the best supervised model (VGG-19), and now show that some hidden layers provide a minor gain in retrieval performance.
> >
> > We hope that these new elements will convince the reviewer that the soundness of our MEG–based image generation is not affected by the choice of layer in the supervised models assessed in Figure 2.
> >
> > **1. Orthogonal objectives**
> >
> > Our decoding pipeline ends with an image generation module conditioned by three pretrained embeddings: CLIP-Vision (last layer), CLIP-Text (last layer) and AutoKL (bottleneck layer). In this context, the retrieval analysis shown in Figure 2 does *not* use supervised embeddings as a baseline for comparing the performance of these three embeddings. Rather, it serves a different purpose: setting up an initial benchmark of a wide family of vision embeddings for MEG-based retrieval. Expanding this benchmark to other layers than the last one would be an important next step but is beyond the scope of this paper.
> >
> > To clarify this issue we will update the manuscript as follows:
> >
> > > “As shown in Figure 2, these embeddings yield a variety of retrieval performances. Future work remains necessary to further probe which layer, or which combination of layers and models may be optimal to retrieve images from brain activity.
> > ”
> >
> > **2. Soundness of the retrieval analysis**
> >
> > It is true that some neuroimaging studies focus on hidden layers of supervised models. Encoding studies (e.g., Schrimpf et al., 2021; Wang et al., 2023; Huth et al 2016) aim to evaluate whether different models are more or less “brain-like”. Therefore, encoding necessitates a fair comparison of each model’s ability to linearly predict brain activity, and thus layer-wise comparison. Decoding, however, aims to maximize retrieval or generation performance. For this, any (combination of) layers and models may legitimately be used, so long as they work (i.e., yield better performance than chance or previous benchmarks).
> >
> > **3. Extension of our analyses to intermediate layers of supervised models**
> >
> > We now replicated our analyses on 4 layers of the best supervised model (VGG-19). The results show that some of these layers slightly outperform the last layer.
> >
> > | Latent                        | top-5 accuracy (%) |
> > |-------------------------------|--------------------|
> > | VGG-19 (last layer)            |         70.3 ± 2.8 |
> > | VGG-19 (avgpool)              |         73.8 ± 2.4 |
> > | VGG-19 (classifier_dropout_2) |         73.8 ± 2.6 |
> > | VGG-19 (classifier_dropout_5) |         74.5 ± 2.4 |
> > | VGG-19 (features_maxpool2d_35)         |         64.3 ± 2.8 |
> > | CLIP-Vision (CLS)               |         63.2 ± 3.0 |
> > | DINOv2 (CLS)                    |         68.0 ± 2.9 |
> >
> > We will update the manuscript with these new analyses applied to the supervised models in the final version of the paper, as these results do not impact the subsequent generation analyses.
> >
> > Thank you for having raised this concern.
> >
> > References
> > ```
> > Schrimpf, Martin, et al. "Artificial neural networks accurately predict language processing in the brain." bioRxiv (2020): 2020-06.
> > Wang, Aria Y., et al. "Better models of human high-level visual cortex emerge from natural language supervision with a large and diverse dataset." Nature Machine Intelligence (2023): 1-12.
> > ```

---

### Official Review · Reviewer_hWdQ · 2023-11-01

**Soundness:** 4 excellent
**Presentation:** 3 good
**Contribution:** 4 excellent
**Rating:** 8
**Confidence:** 4

**Summary:**

In this paper, the authors developed a model based on contrastive and regression objectives to decode MEG, resulting in 7X improvement in image retrieval over a classic linear decoder. The promising results in image retrieval and generation are significant in that the presented approach allows the monitoring of the unfolding of visual processing in the brain based on MEG signals, which have much higher temporal resolution than fMRI. The work yields two potentially interesting observations: (1) late responses are best decoded with DINOv2, and (2) MEG signals contain high-level features, whereas 7T fMRI allows the recovery of low-level features, though it would be worthwhile to articulate or speculate what these findings mean for understanding the cascade of visual processes in the brain.

**Strengths:**

The work is significant in that there is no MEG decoding study that learns end-to-end to reliably generate an open set of images. Thus, it can potentially be considered a ground-breaking in this area of research, even though the techniques used are not necessarily novel from an ML perspective.

**Weaknesses:**

The decoding work is supposed to provide new insights to the cascade of visual processing and the unfolding of visual perception in the brain.  The authors need to articulate better what insights the current observations (mentioned in the Summary) actually provide us.

**Questions:**

What do the two observations tell us about the unfolding of visual perceptual processes?

---

> ### Author Response · Authors · 2023-11-21
>
> We thank the reviewer for their positive assessment of our work and their helpful comment.
>
> We agree that the insights provided by our observations were insufficiently clear. The primary contributions of this study are methodological rather than hypothesis-driven. We have now amended the manuscript to indicate:
>
> > “Our approach provides three main contributions: our MEG decoder (1) yields a 7X increase in performance as compared to linear baselines (Fig. 2), (2) helps reveal when high-level semantic features are processed in the brain (Fig. 3) and (3) allows the continuous generation of images from temporally-resolved brain signals (Fig. 4). Overall, this approach thus paves the way to better understand the unfolding of the brain responses to visual inputs.”
>
> In addition, to clarify and substantiate the insights provided by this study, we have now added several new analyses.
>
> First, we now implement a high-temporally resolved analysis of retrieval and generative decoders, using both sliding and growing windows (Updated figures 3, 4; new figures S6A, S7, S9 ). Overall, these results clarify when low- and high-level representations can be decoded from brain activity, and thus shed lights on the dynamics of visual processing in the brain:
>
> > “Consistent with the decoding peaks observed after image onset and offset (Fig. 3), the retrieval performance of all growing-window models considerably improves after the offset of the image. Together, these results suggest that the brain activity represents both low- and high-level features even after image offset. This finding clarifies mixed results previously reported in the literature. Carlson et al., (2011; 2013), reported small but significant decoding performances after image offset. However, other studies (Cichy et al., 2014, Hebart et al., 2023) did not observe such a phenomenon. In all of these cases, decoders were based on pairwise classification of object categories and on linear classifiers. The improved sensitivity brought by (1) our deep learning architecture, (2) its retrieval objective and (3) its use of pretrained latent features may thus help clarify the dynamics of visual representations in particular at image offset. We speculate that such offset responses could reflect an intricate interplay between low- and high-level processes that may be difficult to detect with a pairwise linear classifier. Overall, we hope that the present methodological contribution will help shine light on this understudied phenomenon.”
>
>
> References:
> ```
> Carlson, Thomas A., et al. "High temporal resolution decoding of object position and category." Journal of vision 11.10 (2011): 9-9.
> Carlson, Thomas, et al. "Representational dynamics of object vision: the first 1000 ms." Journal of vision 13.10 (2013): 1-1.
> Cichy, Radoslaw Martin, Dimitrios Pantazis, and Aude Oliva. "Resolving human object recognition in space and time." Nature neuroscience 17.3 (2014): 455-462.
> Hebart, Martin N., et al. "THINGS-data, a multimodal collection of large-scale datasets for investigating object representations in human brain and behavior." Elife 12 (2023): e82580.
> ```
>
> Second, we inspect the model with a new analysis of its weights (Appendix A.10) and clarify that the decoding time course is different for low- and high-level features:
>
> > “We inspect our decoders to better understand how they use information in the time domain. To do so, we leverage the fact that our architecture preserves the temporal dimension of the input up until the output of its convolutional blocks. This output is then reduced by an affine transformation learned by the temporal aggregation layer (see Section 2.3 and Appendix A.1). Consequently, the weights $w^{agg} \in \mathbb{R}^T$ can reveal on which time steps the models learned to focus. To facilitate inspection, we initialize $w^{agg}$ to zeros before training and plot the mean absolute weights of each model (averaged across seeds).
>
> > The results are presented in Fig. S8. While these weights are close to zero before stimulus onset, they deviate from this baseline after stimulus onset, during the maintenance period and after stimulus offset. Interestingly, and unlike high-level features (e.g., VGG-19, CLIP-Vision), low-level features (e.g., color histogram, AutoKL and DINOv2) have close-to-zero weights in the 0.2-0.5 s interval.
>
> > This result suggests that low-level representations quickly fade away at that moment. Overall, this analysis demonstrates that the models rely on these three time periods to maximize decoding performance, including the early low-level responses ($t=0-0.1$s).“

---

> > ### Author Response · Authors · 2023-11-21
> >
> > Third, we now provide a time-resolved analysis of low- and high-level reconstruction metrics (Appendix 11). These results confirm that the early responses to both image onset and offset are primarily associated with low-level metrics. On the other hand, the decoding of brain responses during the second half of the image presentation interval (0.2-0.5 s) appears to be more related to high-level features.
> >
> > We hope these new experiments and observations clarify the insights that can be obtained with our proposed decoding approach.

---

### Official Review · Reviewer_hpxq · 2023-11-02

**Soundness:** 3 good
**Presentation:** 4 excellent
**Contribution:** 2 fair
**Rating:** 8
**Confidence:** 4

**Summary:**

In this paper the authors propose a method to decode brain activity. The main idea is to train an MEG decoder which maps MEG signals to a feature space which is then used to reconstruct images using a pretrained image generator.

The authors show that MEG decoder which is a DNN leads to 7 times improvement over linear decoders which is a common approach in neuroscience studies. Image generation results suggests that it is possible to reconstruct semantically accurate from MEG activity while low-level details are difficult to reconstruct.

**Strengths:**

1. 7x improvement in decoding accuracy over linear decoders. This is an important result which will encourage neuroscience researchers to use DNNs for decoding MEG/fMRI signals.
2. Clear presentation of methods (Figure 1, Section2).

**Weaknesses:**

1. The reconstruction results are not impressive. Even the best examples shown in Figure 5 often do not have the reconstructions of image of same or related category.  Therefore, the title is misleading as the main contribution of this paper in my opinion is DNN based MEG decoder and retrieval results and is not correctly reflected in the title.
2. The decoder is trained using a combination of two loss functions : MSE loss and CLIP loss (equation 3, line 91). There seems to be no ablation study investigating what is the impact of each loss function in retrieval performance. There is one figure in supplementary material Fig S2 E but I am not sure whether it indicates two terms of CLIP loss or two terms of overall loss (CLIP + MSE).
3. In Line 110 authors mention that they select  lambda by sweeping over {0.0, 0.25, 0.5, 0.75, 1.0} and pick the model whose top-5 accuracy is the highest on the large test. Is the hyperparameter search for lambda done on test data?
4. The claim in the abstract "MEG signals primarily contain high-level visual features" does not have sufficient evidence based on the reconstruction results only. It has been shown in literature (even in Things dataset paper Figure 8) that fMRI responses of early visual cortex (which can decode low-level features) are correlated with MEG responses (Cichy et al. 2014, Hebart et al. 2023) in early time windows. Therefore, a stronger evidence is required to back this claim. A possible explanation why the reconstructions can not recover low-level details might be that temporal aggregration layers leads to suppresion of low-level features which are present in a smaller time-window around 100ms. Another possible explanation is that we are predicting a high-level feature  (DINOv2/CLIP etc.) from MEG which may not need information from low-level features and thus the image generated also lack these details.
5. The main result of the paper is 7x improvement over linear decoders. It is not clear where exactly this result is in the paper. A reader needs to compare results in supplementary and Figure 2 in the main text. Simply adding shaded bar in Figure 2 for linear decoder next to each bars can improve clarity

**Questions:**

Please refer to weaknesses section for points to address in rebuttal.

Overall this paper has some new contributions but authors make some claims which do not have sufficient support in the results. Therefore, my recommendation would be to either tone down the claims or present good evidence to back them up

---

> ### Author Response · Authors · 2023-11-21
>
> We thank our reviewer for their thorough assessment and their helpful comments.
>
> 1. Unimpressive reconstruction.
>
> We agree that the quality of generated images are low in comparison to what can be obtained with 7T fMRI (Table 1).
>
> However, we respectfully disagree that this is not “impressive”. To the best of our knowledge, this is the first study showing MEG-based reconstruction of visual processes. Relative to fMRI, MEG has a very low spatial resolution. Consequently, reconstructing these rich contents is far from obvious – for what it’s worth, we certainly did not expect these results.
>
> We thus clarify our contribution as follows:
>
> > “Our approach provides three main contributions: our MEG decoder (1) yields a 7X increase in performance as compared to linear baselines (Fig. 2), (2) helps reveal when high-level semantic features are processed in the brain (Fig. 3) and (3) allows the continuous generation of images from temporally-resolved brain signals (Fig. 4). Overall, this approach thus paves the way to better understand the unfolding of the brain responses to visual inputs.”
>
> Beyond these subjective considerations, part of this underwhelming feeling may be due to our original figures. We originally opted to automatically select those with the best, average and worst aggregated score across low-level (SSIM) and high-level metrics (SwAV) as there is no standard metric to rank images decoded from the brain. However, after closer inspection, we realize that this SSIM-SwAV combination may not faithfully reflect whether images are accurately reconstructed. To address this issue, we now added two novel figures (Figures 4 and S6, in Appendix A.7), which show manually-selected cases of successful and failed reconstructions, as well as their dynamics. In particular, we now note:
>
> > “Figure S6 shows examples of failed generations. Overall, they appear to encompass different types of failures. Some generations appear to miss the correct category of the true object (e.g., bamboo, batteries, bullets and extinguisher in columns 1-4), but generate images with partially similar textures. Other generations appear to recover some category-level features but generate unrealistic chimeras (bed: weird furniture, alligator: swamp beast; etc, in columns 5-6). Finally, some generations seem to be plain wrong, with little-to-no preservation of low- or high-level features (columns 7-8). We speculate that these different types of failures may be partially resolved with different methods, such as better generation modules (for chimeras) and optimization on both low- and high-level features (for category errors).”
>
> In any case, we agree that these generations are far from being perfect. We thus toned down our abstract by emphasizing the preliminary aspect of our results:
>
> > “Overall, these results, *while preliminary*, provide an important step towards the decoding - in real-time - of the visual processes continuously unfolding within the human brain.”
>
> 2. Impact of $\lambda$ on retrieval.
>
> The reviewer is right that we did not include $\lambda$ from Equation 3 in the hyperparameter search presented in Appendix A.2. As such, Figure S1 presents retrieval performance for models trained with the CLIP loss term only (i.e., with $\lambda=1.0$). We decided not to include $\lambda$ in this hyperparameter search because the focus of this search was on the retrieval task, which is well aligned with the CLIP loss. However, as detailed in the next answer, we did implement a hyperparameter selection procedure for $\lambda$, but that is carried out in a second step, as it relates to a generation-based evaluation.
>
> 3. $\lambda$ search on test data.
>
> The hyperparameter search for $\lambda$ was performed on the “large” test set. The model obtained with the best $\lambda$ was then used for the generation of images only on the “small” test set. Importantly, the large and the small test sets are disjoint, thus there is no leakage. (Of note, we fixed a typo in the original manuscript: the set of $\lambda$ values should in fact have read \{0.0,0.25,0.5,0.75\}.)
>
> We now clarify this issue in the methods:
>
> In Section 2.3:
>
> > "We select $\lambda$ in $\mathcal{L}_{Combined}$ by sweeping over \{0.0,0.25,0.5,0.75\} and pick the model whose top-5 accuracy is the highest on the "large" test set (which is disjoint from the "small" test set used for generation experiments; see Section 2.8)."
>
> In Appendix A.2:
>
> > "We run a hyperparameter grid search to find an appropriate configuration (MEG preprocessing, optimizer, brain module architecture and CLIP loss) for the MEG-to-image retrieval task."
>
> > "We focus the search on the retrieval task, \ie by setting $\lambda=0$ in Eq. 3, and leave the selection of an optimal $\lambda$ to a model-specific sweep using a held-out set (see Section 2.3)."

---

> > ### Author Response · Authors · 2023-11-21
> >
> > 4. Not enough evidence for prevalence of high-level visual features in MEG:
> >
> > We originally indicated that “MEG signals primarily contain high-level visual features" because (1) retrieval scores were the highest with high-level latent embeddings like CLIP-Vision and DINOv2 and (2) because image generation from MEG seems to preserve low-level features much less than fMRI.
> >
> > However, the reviewer is right that these elements are insufficient to claim that MEG *primarily* contains high-level features. To clarify this issue, we (1) added new analyses and (2) amended the abstract and discussion.
> >
> > **New analyses**
> >
> > We provide new analyses to clarify that:
> >
> > > “Together, these results suggest that it is not the reconstruction pipeline which fails to reconstruct low-level features, but rather the MEG signals which are comparatively harder to decode.”
> >
> > First, we increased the temporal resolution of the sliding window analysis by 10X, and added a new growing window analysis (Appendix A.8), which shows that low-level latents (e.g., color histogram) can already be retrieved quickly after image onset, but quickly fade after a few hundreds of milliseconds. By contrast, high-level embeddings (e.g., DINOv2, CLIP-Vision) continue to be retrievable during this late maintenance period.
> >
> > > “Consistent with the decoding peaks observed after image onset and offset (Fig. 3), the retrieval performance of all growing-window models considerably improves after the offset of the image. Together, these results suggest that the brain activity represents both low- and high-level features even after image offset. This finding clarifies mixed results previously reported in the literature. Carlson et al., (2011; 2013), reported small but significant decoding performances after image offset. However, other studies (Cichy et al., 2014, Hebart et al., 2023) did not observe such a phenomenon. In all of these cases, decoders were based on pairwise classification of object categories and on linear classifiers. The improved sensitivity brought by (1) our deep learning architecture, (2) its retrieval objective and (3) its use of pretrained latent features may thus help clarify the dynamics of visual representations in particular at image offset. We speculate that such offset responses could reflect an intricate interplay between low- and high-level processes that may be difficult to detect with a pairwise linear classifier. Overall, we hope that the present methodological contribution will help shine light on this understudied phenomenon.”
> >
> > References
> > ```
> > Carlson, Thomas A., et al. "High temporal resolution decoding of object position and category." Journal of vision 11.10 (2011): 9-9.
> > Carlson, Thomas, et al. "Representational dynamics of object vision: the first 1000 ms." Journal of vision 13.10 (2013): 1-1.
> > Cichy, Radoslaw Martin, Dimitrios Pantazis, and Aude Oliva. "Resolving human object recognition in space and time." Nature neuroscience 17.3 (2014): 455-462.
> > Hebart, Martin N., et al. "THINGS-data, a multimodal collection of large-scale datasets for investigating object representations in human brain and behavior." Elife 12 (2023): e82580.
> > ```
> >
> > Second, following the reviewer’s request, we further inspect whether our decoder focuses on early and/or late brain responses (Appendix A.10):
> >
> > > “We inspect our decoders to better understand how they use information in the time domain. To do so, we leverage the fact that our architecture preserves the temporal dimension of the input up until the output of its convolutional blocks. This output is then reduced by an affine transformation learned by the temporal aggregation layer (see Section 2.3 and Appendix A1). Consequently, the weights $w^{agg} \in \mathbb{R}^T$ can reveal on which time steps the models learned to focus. To facilitate inspection, we initialize $w^{agg}$ to zeros before training and plot the mean absolute weights of each model (averaged across seeds)."
> >
> > > "The results are presented in Fig. S8. While these weights are close to zero before stimulus onset, they deviate from this baseline after stimulus onset, during the maintenance period and after stimulus offset. Interestingly, and unlike high-level features (e.g., VGG-19, CLIP-Vision), low-level features (e.g., color histogram, AutoKL and DINOv2) have close-to-zero weights in the 0.2-0.5 s interval."
> >
> > > "This result suggests that low-level representations quickly fade away at that moment. Overall, this analysis demonstrates that the models rely on these three time periods to maximize decoding performance, including the early low-level responses ($t=$0-0.1 s)."

---

> > > ### Author Response · Authors · 2023-11-21
> > >
> > > (Item 4, continued) Third, we now provide a time-resolved analysis of low- and high-level reconstruction metrics (Appendix A.11). These results confirm that the early responses to both image onset and offset are primarily associated with low-level metrics. On the other hand, the decoding of brain responses during the second half of the image presentation interval (0.2-0.5 s) appears to be more related to high-level features.
> > >
> > > Overall, these elements confirm and extend previous findings (Cichy et al., 2014; Hebart et al., 2023) showing that:
> > > low-level representations are already present in the early MEG responses, and
> > > high-level representations are specifically maintained and integrated in the late brain responses.
> > >
> > > **Amended paragraphs**
> > >
> > > Abstract:
> > >
> > > > “Third, image retrievals and generations both suggest that high-level visual features can be decoded from MEG signals, although the same approach applied to 7T fMRI also recovers better low-level features.”
> > >
> > > Results:
> > >
> > > > “As confirmed by the evaluation metrics of Table 1 (see Table S3 for participant-wise metrics), many generated images preserve the high-level category of the true image. However, most generations appear to preserve a relatively small amount of low-level features, such as the position and color of each object.”
> > >
> > > > “Together, these results suggest that it is not the reconstruction pipeline which fails to reconstruct low-level features, but rather the MEG signals which are comparatively harder to decode.
> > >
> > > Discussion:
> > >
> > > > “First, generating images from MEG appears worse at preserving low-level features than a similar pipeline on 7T fMRI (Fig. S5). This result resonates with the fact that the spatial resolution of MEG ($\approx$\,cm) is much lower than 7T fMRI's ($\approx$\,mm). Moreover, and consistent with previous findings (Cichy et al., 2014; Hebart et al., 2023), the low-level features can be predominantly extracted from the brief time windows immediately surrounding the onset and offset of brain responses. As a result, these transient low-level features might have a lesser impact on image generation compared to the more persistent high-level features.”
> > >
> > > 5. Thank you for the suggestion. We have updated Figure 2 accordingly by including baseline linear decoder results directly in the figure.

---

> > > > ### Comment · Reviewer_hpxq · 2023-11-21
> > > > **Thanks for clarification and additional results**
> > > >
> > > > I would like to thank authors for their comprehensive responses backed by new results. I have updated my rating to 8.

---

### Official Review · Reviewer_4PGh · 2023-11-02

**Soundness:** 3 good
**Presentation:** 3 good
**Contribution:** 2 fair
**Rating:** 6
**Confidence:** 3

**Summary:**

This contribution concerns the interesting topic of decoding/retrieval and reconstructing of visual input from MEG data (THINGS-MEG data set). The approach is based on representations of images and MEG data using multiple architectures and multiple levels of generalization.

There is a rich literature on decoding and reconstructing visual and audio stimulus from brain recordings, so novelty is somewhat limited.

Decoding is evaluated as retrieval in closed and open set conditions (the latter using zero-shot setting).
Retrieval is based on linking by learning to align MEG and image representations

The reconstruction of visual input is based on generative models, using frameworks that have been developed elsewhere (Ozcelik and Van Rullen).

Compared to the very rich literature on methods based on MEG and other modalities, this study has an increased focus on temporal resolution of the retrieval process and furthermore, they use diffusion models for conditional generation.

**Strengths:**

Compared to the very rich literature on methods based on MEG and other modalities, this study has increased focus on temporal resolution of the retrieval and furthermore using sota diffusion models for conditional generation.
It is concluded that retrieval interesting peaks following image onset and image offset (the latter based on the after-image presumably). Retrieval performance is good for several image representations (VGG and DINOv2)
The generative performance is evaluated in a number of metrics, there is good consistency among the metrics.
Visually the generation makes sense.
Useful to see examples  stratified over good, bad and ugly cases.

**Weaknesses:**

There is a rich literature on decoding and reconstructing visual and audio stimulus from brain recordings, so novelty is somewhat limited.

Based on MEG we have high time resolution and SNR. In the temporally resolved analysis, it is interesting that VGG outperforms the more advanced representations for the direct image (after image onset) while the more complex image representations dominate retrieval based on the after-image (following image offset). We miss a discussion of this interesting finding.

The generative performance is evaluated in a number of metrics with good consistency among the metrics. Yet, we are missing uncertainty estimates to weigh the evidence in this case

Visually the generated imagery is intriguing. However, we miss a discussion of the notable lack of fine grained semantic relatedness (generation seems primarily to pick up on texture, object scale(?) and high-level semantics eg. man-made vs natural)

**Questions:**

Based on MEG we have high time resolution and SNR. In the temporally resolved analysis, it is interesting that VGG outperforms the more advanced for the direct image (after onset) while the more complex image representations dominate retrieval based on the after image (after image offset). Missing a discussion of this interesting finding.

The generative performance while evaluated in a number of metrics with good consistency among the metrics. Yet, we are missing uncertainty estimates to weigh the evidence in this case

Visually the generated imagery is intriguing. However, we miss a discussion of the notable lack of fine grained semantic relatedness (generation seems primarily to pick up on texture, object scale(?) and high-level semantics eg. man-made vs natural)

---

> ### Author Response · Authors · 2023-11-21
>
> We thank the reviewer for their positive assessment of our work and their insightful comments. We address every comment and question below and highlight what we changed in the manuscript as a result.
>
> 1. We agree that the *conceptual* novelty is limited: there is a lot of work on the decoding of images from brain activity. However, our method and our results are novel: Past research has focused on either (1) classification (Grootswagers et al., 2019; King & Wyart, 2021) (2) encoding (Cichy et al., 2017; Gifford et al., 2022) tasks from M/EEG or (3) image reconstruction from fMRI (Seeliger et al., 2018; VanRullen & Reddy, 2019; Ozcelik & VanRullen, 2023). To our knowledge, there is no study on image reconstruction from MEG, and thus no model or benchmark to reconstruct natural images from the unfolding of brain activity. We now clarify this issue in the discussion:
>
> > “Our approach provides three main contributions: our MEG decoder (1) yields a 7X increase in performance as compared to linear baselines (Fig. 2), (2) helps reveal when high-level semantic features are processed in the brain (Fig. 3) and (3) allows the continuous generation of images from temporally-resolved brain signals (Fig. 4). Overall, this approach thus paves the way to better understand the unfolding of the brain responses to visual inputs.“
>
> To emphasize this point, we updated Figures 3 and 4 and added Appendix A.8, which now show a highly-resolved decoding of MEG responses, based on retrieval and generation with sliding or growing window decoders.
>
> 2. We agree that it is interesting to highlight the differences between VGG and other high-level embedding observed after image onset and image offset. This is carried out in Appendix A.11.
>
> > “Lastly, we provide a sliding window analysis of these metrics in Appendix A.11. These results suggest that early responses to both image onset and offset are primarily associated with low-level metrics, while high-level features appear more related to brain activity in the 200-500 ms interval."
>
> 3. The reviewer is right that we forgot to add uncertainty estimates. We now added SEM (standard error of the mean) for each relevant metric. Finally, we included a new table in Appendix 9 reporting individual metrics and SEM for each participant (Table S3).

---

> > ### Author Response · Authors · 2023-11-21
> >
> > 4. We agree that we miss an additional discussion about the lack of fine-grained semantic relatedness. To address this issue we now performed two new analyses.
> >
> > First, we showcase and briefly discuss different types of failed reconstructions shown in the new Figure S6 in Appendix A.7:
> >
> > > “Figure S6 shows examples of failed generations. Overall, they appear to encompass different types of failures. Some generations appear to miss the correct category of the true object (e.g., bamboo, batteries, bullets and extinguisher in columns 1-4), but generate images with partially similar textures. Other generations appear to recover some category-level features but generate unrealistic chimeras (bed: weird furniture, alligator: swamp beast; etc, in columns 5-6). Finally, some generations seem to be plain wrong, with little-to-no preservation of low- or high-level features (columns 7-8). We speculate that these different types of failures may be partially resolved with different methods, such as better generation modules (for chimeras) and optimization on both low- and high-level features (for category errors).”
> >
> > Second, we compute the generation metrics as a function of time (Figure S9). These results show that the early onset and early offset responses are decodable primarily through low-level features. Together with our updated sliding window and growing window decoding analyses, these results strengthen and extend previous findings. We now indicate:
> >
> > > “Consistent with the decoding peaks observed after image onset and offset (Fig. 3), the retrieval performance of all growing-window models considerably improves after the offset of the image. Together, these results suggest that the brain activity represents both low- and high-level features even after image offset. This finding clarifies mixed results previously reported in the literature. Carlson et al., (2011; 2013), reported small but significant decoding performances after image offset. However, other studies (Cichy et al., 2014; Hebart et al., 2023) did not observe such a phenomenon. In all these cases, decoders were based on pairwise classification of object categories and on linear classifiers. The improved sensitivity brought by (1) our deep learning architecture, (2) its retrieval objective and (3) its use of pretrained latent features may thus help clarify the dynamics of visual representations in particular at image offset. We speculate that such offset responses could reflect an intricate interplay between low- and high-level processes that may be difficult to detect with a pairwise linear classifier. We hope that the present methodological contribution will help shine light on this understudied phenomenon.”
> >
> > References:
> > ```
> > Carlson, Thomas A., et al. "High temporal resolution decoding of object position and category." Journal of vision 11.10 (2011): 9-9.
> > Carlson, Thomas, et al. "Representational dynamics of object vision: the first 1000 ms." Journal of vision 13.10 (2013): 1-1.
> > Cichy, Radoslaw Martin, Dimitrios Pantazis, and Aude Oliva. "Resolving human object recognition in space and time." Nature neuroscience 17.3 (2014): 455-462.
> > Hebart, Martin N., et al. "THINGS-data, a multimodal collection of large-scale datasets for investigating object representations in human brain and behavior." Elife 12 (2023): e82580.
> > ```

---

> > ### Comment · Reviewer_4PGh · 2023-11-21
> > **Appendix missing?**
> >
> > I thank the authors for the many last minute comments and clarifications. However, the version I can see in Openreview does not show appendices?

---

> > > ### Author Response · Authors · 2023-11-21
> > >
> > > Thank you for your feedback. Indeed, authors are asked to upload appendices as a separate PDF file, which can be downloaded by clicking on the 'pdf' link on the right of 'Supplementary material', immediately below the Abstract on the submission page.
> > >
> > > Here is also the [link](https://openreview.net/attachment?id=3y1K6buO8c&name=supplementary_material) for convenience.
> > >
> > > Please don’t hesitate to ping us if this link does not work.

---

> > > > ### Author Response · Authors · 2023-11-22
> > > >
> > > > Dear reviewer 4PGh,
> > > >
> > > > We appreciate your thoughtful feedback and have made careful changes to our manuscript based on your comments. We believe we have now addressed all of them:
> > > >
> > > > 1. We further supported the novelty of our results;
> > > > 2. Provided additional comparisons between the different image representations;
> > > > 3. Included uncertainty estimates and showed additional generation results, including new time-resolved generations.
> > > >
> > > > We would be grateful if you could confirm whether the revisions meet your expectations. If you have any further questions or concerns, please don't hesitate to reach out to us for additional discussion.
> > > >
> > > > Thank you.

---

> > > > > ### Comment · Reviewer_4PGh · 2023-11-23
> > > > > **Upgrade**
> > > > >
> > > > > While I am still unsure about the contribution for ICLR I acknowledge the many improvements made and therefore change my grade to 6

---

### Author Response · Authors · 2023-11-21
**General answer**

We are thankful to our four reviewers for their thorough feedback.

Overall, the paper was **evaluated positively on all three dimensions**:
* Soundness: from “good” to “excellent”
* Presentation: from “good” to “excellent”
* Contribution: from “fair” to “excellent”

Reviewers confirmed that ‘there is no MEG decoding study that learns end-to-end to reliably generate an open set of images’ (hWdQ), although this study extends existing efforts in fMRI. The work was considered “important” (Hpxq), “significant” (‘hWdQ’) and “potentially [...] ground-breaking in this area of research” (hWdQ).

The reviewers **did not express any major technical concerns**. However, two main perspective elements were raised with regard to our contribution. First, the accuracy of the reconstructions led to mixed reviews. For example,
* \+ 4PGh: “visually the generation makes sense”
* \+ hWdQ: ” The promising results in image retrieval and generation are significant”
* \+ F12m: “MEG decoder leads to reasonable image retrieval and image generation”
* \- hpxq: “the reconstruction results are not impressive”.

We believe that this mixed reaction is partly due to a suboptimal presentation of our results. Consequently, we now showcase clear success and failure modes over time.

Second, several reviewers recommended **discussing further the neuroscientific insights**, e.g.,:
* hpxq: “it would be worthwhile to articulate or speculate what these findings mean for understanding the cascade of visual processes in the brain.”
* 4PGh: “we miss a discussion of this interesting finding [decoding at image offset]”;
* F12m: “As with other deep decoding papers, it is not super clear what the ultimate scientific insight is. This is not a criticism specific to this paper, but more generally aimed at current decoding works [...] “it would benefit the paper to add some discussion on [...] scientific insights”
* hWdQ: “it would be worthwhile to articulate or speculate what these findings mean for understanding the cascade of visual processes in the brain.”

We believe ICLR is a suitable place for strictly methodological/result achievements: namely here, the **first demonstration of image generation from temporally-resolved signals**, with a 7X improvement over a standard linear baseline adapted from SOTA in fMRI (Ozcelik & VanRullen, 2023). This technical development potentially unlocks future research where time is critical (e.g., video decoding).

However, we do agree with the overall concern: unlike encoding, decoding is notoriously difficult to use for neuroscientific theory (e.g., Naseralis et al., 2011; DiCarlo et al., 2007). Consequently, we added a **series of new analyses dedicated to a neuroscience audience**:
* A much more fine-grained temporal resolution of image retrievals and generations (Figures 3, S7);
* Examples of time-resolved image generation (Figures 4, S6);
* An analysis of the temporal weights of the decoder (Figure S8);
* A time-resolved evaluation of low- and high-level metrics (Figure S9).

Overall, these new analyses reveal that (1) late brain responses specifically represent high-level features and (2) the brain responses to image offset represent both low- and high-level responses. These elements subsume previous neuroscientific investigation.

These new analyses show that unlike low-level features, high-level features are clearly maintained in the late period of the MEG responses (extending previous findings, e.g., Cichy et al., (2014); Hebart et al., (2023)). In addition, these analyses reveal the representations elicited by the image offset. We now discuss how these technical developments help understand the unfolding and maintenance of visual processing in the brain.

Finally, we retrained generation models as we noticed a discrepancy between the reported hyperparameters and actual models used in the original submission. With the new models, image reconstruction metrics improve marginally in Table 1. Our original conclusions are not affected.

Overall, we hope that these updated results and discussion will strengthen our original contributions.

---

> ### Author Response · Authors · 2023-11-21
> **Update to supplementary materials**
>
> We have now uploaded a “diff” of the pdf with changes highlighted in blue in the Supplementary Materials zip file. Of note, the zip file also contains a pdf of the appendices. We would like to thank the reviewers again for their time and feedback.

---

### Meta-Review · Area_Chair_tzEx · 2023-12-10

**Metareview:**

The paper describes a decoding pipeline for re-constructing the visual input a human observer sees from MEG real-time recordings. The work is inspired by previous approaches using fMRI recordings and their alignments pretrained ML embeddings but contains idiosyncratic adaptations taking into account the characteristic differences between fMRI and MEG. Using MEG signals for reconstructing visual input seems novel and innovative. The time-dependent decoding analysis which is possible with MEG signals provides insights into the dynamics of the neurobiology of visual processing. The experimental demonstration of the method is well described in the paper and convincing.

Reviewers commonly acknowledged the novelty of using MEG for decoding. The also seem to largely agree with the details of implementation although some details needed to be discussed in an overall lively author/reviewer discussion. As a result of the discussion, two reviewer increased their scores. The novelty of the approach to MEG signals and the potential insights the methods can provide in terms of the dynamics of visual processing in the human visual system (see Fig. 4) stand out.

**Justification For Why Not Higher Score:**

While the application to MEG signals is novel, the technical aspects of the decoding pipeline largely follow previous studies using fMRI signals, thus technical innovation is limited.

**Justification For Why Not Lower Score:**

The novelty of using MEG signal for reconstructing in-real time the dynamics of visual processing in the human visual system is high and the application promises lot of potential. In particular, the intrinsic advantage in temporal resolution in using MEG compared to fMRI signals potentially can provide very interesting insight in the dynamics of biological visual processing.

---

### Decision · Program_Chairs · 2024-01-16

Accept (poster)